# Random Erasing vs. Model Inversion: A Promising Defense or a False Hope?

**Viet-Hung Tran**[*]                                                              *h.tran@qub.ac.uk*
*Singapore University of Technology and Design (SUTD); The Queen's University Belfast*

**Ngoc-Bao Nguyen**[*]                                                *thibaongoc_nguyen@sutd.edu.sg*
*Singapore University of Technology and Design (SUTD)*

**Son T. Mai**                                                          *thaison.mai@qub.ac.uk*
*The Queen's University Belfast*

**Hans Vandierendonck**                                           *h.vandierendonck@qub.ac.uk*
*The Queen's University Belfast*

**Ira Assent**                                                              *ira@cs.au.dk*
*Aarhus University*

**Alex Kot**                                                              *eackot@ntu.edu.sg*
*Nanyang Technological University (NTU)*

**Ngai-Man Cheung**[†]                                            *ngaiman_cheung@sutd.edu.sg*
*Singapore University of Technology and Design (SUTD)*

**Reviewed on OpenReview:** *https://openreview.net/forum?id=S9CwKnPHaO*

## Abstract

Model Inversion (MI) attacks pose a significant privacy threat by reconstructing private training data from machine learning models. While existing defenses primarily concentrate on model-centric approaches, the impact of data on MI robustness remains largely unexplored. In this work, we explore *Random Erasing (RE)*—a technique traditionally used for improving model generalization under occlusion—and uncover its surprising effectiveness as a defense against MI attacks.

Specifically, our novel feature space analysis shows that model trained with RE-images introduces a significant discrepancy between the features of MI-reconstructed images and those of the private data. At the same time, features of private images remain distinct from other classes and well-separated from different classification regions. These effects collectively degrade MI reconstruction quality and attack accuracy while maintaining reasonable natural accuracy. Furthermore, we explore two critical properties of RE including Partial Erasure and Random Location. First, *Partial Erasure* prevents the model from observing entire objects during training, and we find that this has significant impact on MI, which aims to reconstruct the entire objects. Second, the *Random Location* of erasure plays a crucial role in achieving a strong privacy-utility trade-off. Our findings highlight RE as a simple yet effective defense mechanism that can be easily integrated with existing privacy-preserving techniques. Extensive experiments of 37 setups demonstrate that our method achieves SOTA performance in privacy-utility tradeoff. The results consistently demonstrate the superiority of our defense over existing defenses across different MI attacks, network architectures, and attack configurations. For the first time, we achieve significant degrade in attack accuracy *without* decrease in utility for some configurations. Our code and additional results are available at: `https://ngoc-nguyen-0.github.io/MIDRE/`

---

[*] The first two authors contributed equally. [†] Corresponding author.

# 1 Introduction

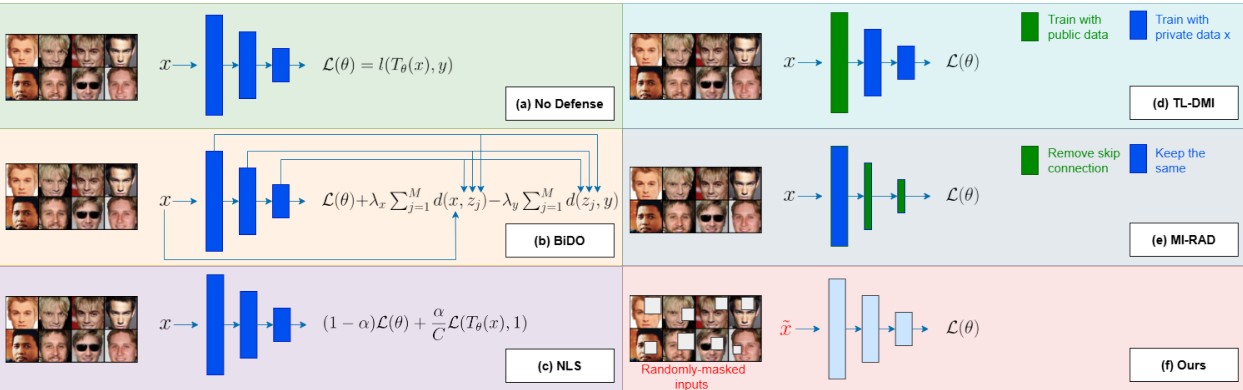

Figure 1: **Our Proposed Model Inversion (MI) Defense via Random Erasing (MIDRE).** (a) "No Defense": Training a model without MI defense. $\mathcal{L}(\theta)$ is the standard training loss, e.g., cross-entropy. Training a model with state-of-the-art MI defense (SOTA) (b) BiDO (Peng et al., 2022), (c) NLS (Struppek et al., 2024), and (d) TL-DMI (Ho et al., 2024), (e) MI-RAD (Koh et al., 2024) , (f) Our method. Studies in (Peng et al., 2022; Struppek et al., 2024) focus on **adding new loss** to the training objective in other to find the balance between model utility and privacy. Both TL-DMI (Ho et al., 2024) and MI-RAD (Koh et al., 2024) focus on **the model's parameters** to defend against MI. For our proposed method (f), the training procedure and objective are the same as that in (a) "No Defense". However, the training samples presented to the model are partially masked, thus, reducing private training sample's information encoded in the model and **preventing the model from observing the entire images**. Therefore, **MIDRE is different from other approaches and focuses on input data only to defend.** We find that this can significantly degrade MI attacks, which require substantial amount of private training data information encoded inside the model in order to reconstruct high-dimensional private images.

Machine learning and deep neural networks (DNNs) (LeCun et al., 2015) have demonstrated their utility across numerous domains, including computer vision (Voulodimos et al., 2018; O'Mahony et al., 2020), natural language processing (Otter et al., 2020), and speech recognition (Deng et al., 2013; Nassif et al., 2019). DNNs are now applied in critical areas such as medical diagnosis (Azad et al., 2021), medical imaging (Shen et al., 2017; Lundervold & Lundervold, 2019), facial recognition (Wang & Deng, 2021; Guo & Zhang, 2019; Masi et al., 2018), and surveillance (Zhou et al., 2021; Harikrishnan et al., 2019; Hashmi et al., 2021). However, the potential risks associated with the widespread deployment of DNNs raise significant concerns. In many practical applications, privacy violations involving DNNs can result in the leakage of sensitive and private data, eroding public trust in these applications. Defending against privacy violations of DNNs is of paramount importance.

One specific type of privacy violation is Model Inversion (MI) attacks on machine learning and DNN models. MI attacks aim to reconstruct private training data by exploiting access to machine learning models. Recent advancements in MI attacks including GMI (Zhang et al., 2020), KedMI (Chen et al., 2021), PPA (Struppek et al., 2022), MIRROR (An et al., 2022), IF-GMI (Qiu et al., 2024), PPDG-MI (Peng et al., 2024), PLG-MI (Yuan et al., 2023), and LOMMA (Nguyen et al., 2023) have achieved remarkable progress in attacking important face recognition models. This raises privacy concerns for models that are trained on sensitive data, such as face recognition, surveillance and medical diagnosis.

**Related works.** Existing MI defenses primarily focus on model-centric strategies like model gradients (Dwork, 2006; 2008), loss functions (Wang et al., 2021; Peng et al., 2022; Struppek et al., 2024), model features (Ho et al., 2024), and architecture designs (Koh et al., 2024) (see Fig. 1). Earlier works (Dwork, 2006; 2008) demonstrated the ineffectiveness of traditional Differential Privacy (DP) mechanisms against Model Inversion (MI) attacks. Recent research (Wang et al., 2021; Peng et al., 2022; Struppek et al., 2024) has explored the impact of loss functions on MI resilience. Wang et al. (2021) restricted the dependency between model inputs and outputs, while BiDO (Peng et al., 2022) focused on limiting the dependency

between model inputs and latent representations. To partially restore model utility, BiDO maximized the dependency between latent representations and outputs. Struppek et al. (2024) proposed using negative label smoothing factors as a defense. However, these loss function-based approaches often introduce conflicting objectives, leading to significant degradation in model utility. Recently, TL-DMI (Ho et al., 2024) restricted the number of layers to be encoded by the private training data, while MI-RAD (Koh et al., 2024) found that removing skip connections in final layers enhances robustness. However, both approaches experience difficulty in achieving competitive balance between utility and privacy.

Data augmentation, a technique that creates new, synthetic samples from existing data points, offers a promising avenue for enhancing model robustness. In this paper, we pioneer the investigation of Random Erasing (RE) (Zhong et al., 2020) for MI defense. RE, traditionally used to improve model generalization for detecting occluded objects by removing randomly a region in training samples, demonstrates its effectiveness as a powerful defense against MI attacks. In MI attacks, adversaries optimize reconstructed images to align with the target model's feature space representation of training samples. As will be shown in our novel analysis, thanks to RE, the target model's feature representations are inherently biased towards the RE-private images, the training data, rather than the private data. Consequently, **RE creates a discrepancy between the features of MI-reconstructed images and that of private images**, resulting in degraded MI attacks. Meanwhile, features of private images remain distinct from other classes, maintaining reasonable natural accuracy. Furthermore, we highlight **two crucial properties of RE that serve as an effective MI defense: *Partial Erasure* and *Random Location***. On the one hand, *Partial Erasure significantly reduces the amount of private information embedded in the training data, preventing the model from observing the entire image*, and consequently degrades the MI attacks. On the other hand, *Random Location improves the diversity of training data*, thereby, enhances the model utility. Our proposed method leads to substantial degradation in MI reconstruction quality and attack accuracy (See Sec. 3 for our comprehensive analysis and validation). Meanwhile, natural accuracy of the model is only moderately affected. Overall, we can achieve state-of-the-art performance in privacy-utility trade-offs as demonstrated in our extensive experiments of 37 setups – 9 SOTA MI attacks including both white-box, black-box, and label-only MI attacks, 11 model architectures (including vision transformer), 6 datasets and different resolution including $64 \times 64$, $160 \times 160$, and $224 \times 224$ resolution – and user study (in Supp.). Our contributions are:

- Our novel defense method, Model Inversion Defense via Random Erasing (MIDRE), is the first work to consider the well-known RE technique as a privacy protection mechanism, leveraging its powerful ability to reduce MI attack accuracy while maintaining model utility. All results support the SOTA effectiveness of a simple technique in addressing a critical security concern.

- We provide a deep understanding on feature space analysis of Random Erasing's defense effectiveness which leads to reduce of MI attacks in MIDRE model.

- Our analysis investigates two crucial properties of RE that serve as an effective MI defense: Partial Erasure and Random Location. With these two properties, our defense method degrades the attack accuracy while the impact on model utility is small.

- We conduct extensive experiments (Sec. 4, Supp.) and user study (Supp.) to demonstrate that our MIDRE can achieve SOTA privacy-utility trade-offs. Notably, in the high-resolution setting, our MIDRE is the first to achieve competitive MI robustness without sacrificing natural accuracy. Note that our method is very simple to implement and is complementary to existing MI defense methods.

## 2 Our Approach: Model Inversion Defense via Random Erasing (MIDRE)

### 2.1 Model inversion

A model inversion (MI) attack aims to reconstruct private training data from a trained machine learning model. The model under attack is called a *target model*, $T_\theta$. The target model $T_\theta$ is trained on a private dataset $\mathcal{D}_{priv} = \{(x_i, y_i)\}_{i=1}^N$, where $x_i$ represents the private, sensitive data and $y_i$ represents the corresponding ground truth label. For example, $T_\theta$ could be a face recognition model, and $x_i$ is a face image of

an identity. The model is trained with standard loss function $\ell$ that penalizes the difference between model output $T_\theta(x)$ and $y$.

**MI attacks.** The underlying idea of MI attacks is to seek a reconstruction $x$ that achieves maximum likelihood for a label $y$ under $T_\theta$:

$$\max_x \mathcal{P}(y; x, T_\theta) \tag{1}$$

In addition, some prior to improve reconstructed image quality can be included (Zhang et al., 2020; Chen et al., 2021). SOTA MI attacks (Zhang et al., 2020; Chen et al., 2021; Nguyen et al., 2023; Struppek et al., 2022) also apply GAN trained on a public dataset $\mathcal{D}_{pub}$ to limit the search space for $x$. $\mathcal{D}_{pub}$ has no identity intersection with $\mathcal{D}_{priv}$, assuming attackers can not access to any private samples.

## 2.2 Random Erasing (RE) as a defense

Random Erasing (RE) (Zhong et al., 2020) involves employing a random selection process to identify an region inside an image. Subsequently, this region is altered through the application of designated pixel values, such as zero or the mean value obtained from the dataset, resulting in *partial masking* of the image. Traditionally, RE is applied as a data augmentation technique to improve robustness of machine learning models in the presence of object occlusion (Zhong et al., 2020).

We propose a simple configuration of RE as a MI defense, requiring only one hyper-parameter. Given a training sample $x$ with dimensions $W \times H$, we propose a square region erasing strategy to restrict private information leakage from $x$. We initiate by randomly selecting a starting point, denoted as $(x_e, y_e)$, within the bounds of $x$. Next, we randomly select the erased area portion $a_e$ within the specified range of $[0.1, a_h]$, guaranteeing at least 10% of $x$ is erased during training, while $a_h$ is the only hyper-parameter of our defense. The size of the erased region is $\sqrt{s_{RE}} \times \sqrt{s_{RE}}$ where $s_{RE} = W \times H \times a_e$ is the erased region. With the designated region, we determine the coordinates of the erased region $(x_e, y_e, x_e + \sqrt{s_{RE}}, y_e + \sqrt{s_{RE}})$. However, we need to ensure this selected region stays entirely within the boundaries of $x$, i.e. $x_e + \sqrt{s_{RE}} \leq W$, $y_e + \sqrt{s_{RE}} \leq H$. If the erased region extends beyond the image width or height, we simply repeat the selection process until we find a suitable square erased region that fits perfectly within $x$. We fill the erased regions with ImageNet mean pixel value (See Supp for a detailed discussion on the impact of the erased value) to obtain the RE-image. Note that RE is applied to all private training samples and the size and position vary each training iteration. We depict our method in Algorithm 1 (Supp.)

# 3 Analysis of privacy effect of MIDRE

In this section, we analyze the privacy impact of RE within our proposed MIDRE framework. We conduct a thorough analysis and demonstrate that RE can achieve a surprisingly state-of-the-art balance between utility and privacy. Specifically, when employed as a defense against MI attacks, RE is the first method to significantly reduce attack accuracy without compromising utility in certain configurations, whereas all prior MI defenses exhibit noticeable degradation in utility to achieve similar reductions in attack success. Experimental results in Sec. 4 further validate this finding.

We delve deeper into the mechanisms that underpin the effectiveness of RE. Importantly, we conduct a feature space analysis to explain RE's defense effectiveness, showing that model trained with MIDRE leads to a discrepancy between the features of MI-reconstructed images and that of private images, resulting in degrading of attack accuracy. At the same time, private images remain distinct from other classes and well-separated from different classification regions, maintaining reasonable natural accuracy. Furthermore, our analysis reveals that partial erasure, as implemented in RE, is a highly effective method for mitigating MI attacks. Particularly, to present the model with less private pixels during training, our approach of applying partial erasure while maintaining the original number of training epochs proves to be more effective than the alternative approach of reducing the number of epochs without using partial erasure. We attribute this to the fact that MI attacks rely on the target model to reconstruct the **entire image**, and RE's partial erasure prevents the target model from ever fully observing the entire image throughout the training process. Additionally, we show that applying partial erasure at random locations, as is done in RE, is more effective than erasure at fixed locations.

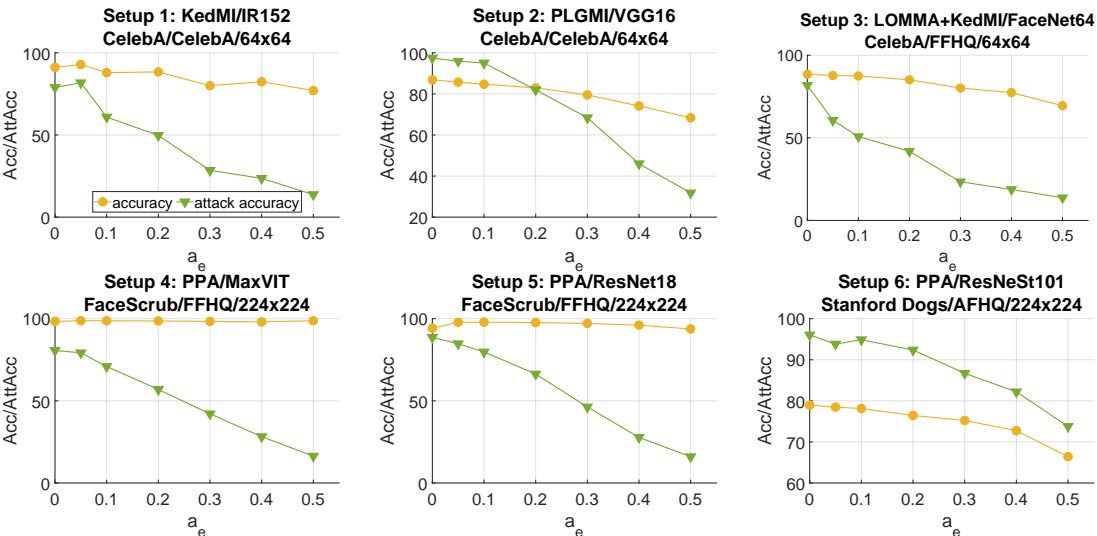

Figure 2: **Our analysis shows that Random Erasing (RE) can lead to substantial degradation in MI reconstruction quality and attack accuracy, while natural accuracy of the model is only moderately affected.** In this analysis, we conduct 6 experimental setups with different *MI attacks/target models architecture/private/public datasets/image resolution*. We analyze the attack (green line) and natural accuracy (orange line) of the target models under different extents of random erasing applied in the training stage, using random erasing ratio $a_e = s_{RE}/(W \times H)$ as discussed in Sec. 2.2. To properly reconstruct private high-dimensional facial images of individuals, MI attacks require significant amount of private training data information encoded inside the model. We find that the model using RE by small percentages can significantly degrade MI attacks, with MI attack accuracy decreasing, for example, from 15.47% to 39.93%. However, the natural accuracy of the model only decreases slightly, less than 4%, as sufficient information remained in the RE-images for the model to learn to discriminate between individuals (Setup 1-3). We also observe a high degradation in MI attack accuracy while the model accuracy increases. For instance, model accuracy increases by 0.37%, while attack accuracy decreases by 69.39% (Setup 4). Overall, our defense demonstrates state-of-the-art privacy-utility trade-offs and can improve model utility in certain setups

## 3.1 RE degrades MI significantly, achieving SOTA privacy-utility trade-off

In the analysis, we study attack accuracy and natural accuracy of a target model $T_\theta$ under different erased region portions $a_e$. Recall $a_e = s_{RE}/(W \times H)$, and $\sqrt{s_{RE}} \times \sqrt{s_{RE}}$ is the dimension of the erased region. For the target model, which is a face recognition model, in each setup, we employ the same architecture and hyper-parameters, while modifying the erased region portions $a_e$. Specifically, we fix the values of $a_e$ for this analysis, to study the effect of $a_e$ to model utility (accuracy) and model privacy (attack accuracy). We vary $a_e$ from 0.0 (indicating no random erasing and the same as No Defense) to 0.5 (erasing 50% of each input image). After the training of $T_\theta$, we proceed to evaluate its top 1 attack accuracy using SOTA MI attacks. This evaluation is conducted for all target models trained with different $a_e$. In order to ensure diversity in our study, we employ six distinct setups for model inversion attacks, target model architecture, private dataset, and public dataset, and both low- and high-resolution datasets.

**RE has small impact on model utility while degrading MI attacks significantly.** Fig. 2 summarizes the impact of erased portions on model performance and model inversion attacks. In all setups, we demonstrably improve robustness against MI attacks with small sacrifice to natural accuracy. For instance, introducing erased portions $a_e$ at a ratio of 0.2 in Setup 1 caused a small 2.76% decrease in natural accuracy while the attack accuracy plummeted by 29.2%. This trend continues in Setup 2 – a 0.2 ratio of $a_e$ led to a modest 3.92% decrease in natural accuracy, but a substantial 15.47% drop in attack accuracy. We note that in Setup 3, LOMMA+KedMI attack accuracy degrades by 39.93%. For high resolution images (Setup 4, 5), we observe an increase in model accuracy when using RE. In Setup 4, there is a significant 69.39% drop in attack accuracy while natural accuracy slightly increase (0.37%) when $a_e = 0.5$. Similar trend for Setup 5,

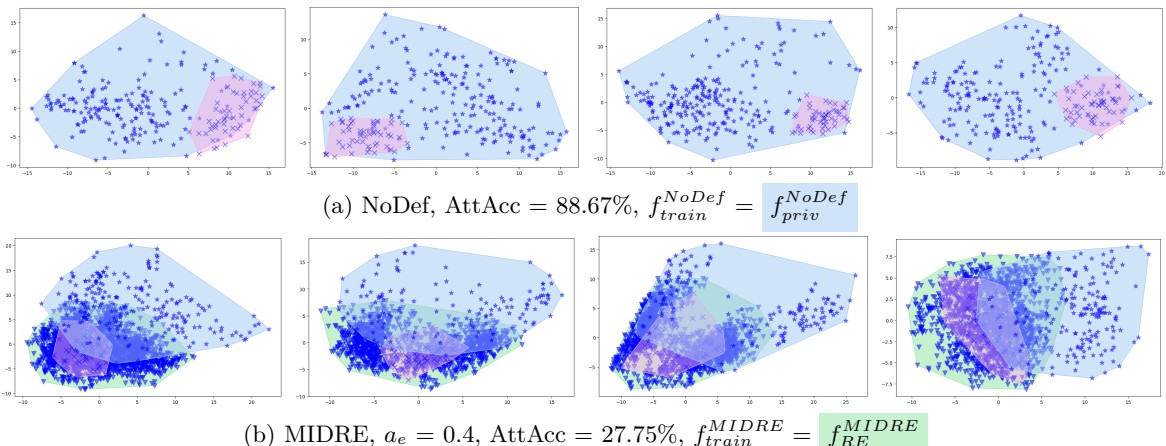

(a) NoDef, AttAcc = 88.67%, $f_{train}^{NoDef} = f_{priv}^{NoDef}$

(b) MIDRE, $a_e = 0.4$, AttAcc = 27.75%, $f_{train}^{MIDRE} = f_{RE}^{MIDRE}$

Figure 3: **Feature space analysis to show that, under MIDRE, $f_{recon}^{MIDRE}$ and $f_{priv}^{MIDRE}$ has a discrepancy, degrading MI attack.** We visualize penultimate layer activations of private images ($\star$ $f_{priv}$), RE-private images ($\blacktriangledown$ $f_{RE}$), and MI-reconstructed images ($\times$ $f_{recon}$) generated by both (a) NoDef and (b) our MIDRE model. We also visualize the convex hull for private images , RE-private images , and MI-reconstructed images . In (a), $f_{recon}^{NoDef}$ closely resemble $f_{priv}^{NoDef}$, consistent with high attack accuracy. In (b), private images and RE-private images share some similarity but they are not identical, with partial overlap between $f_{priv}^{MIDRE}$ and $f_{RE}^{MIDRE}$ . Importantly, $f_{recon}^{MIDRE}$ closely resembles $f_{RE}^{MIDRE}$ as RE-private is the training data for MIDRE. This results in **a reduced overlap between $f_{recon}^{MIDRE}$ and $f_{priv}^{MIDRE}$ , explaining that MI does not accurately capture the private image features under MIDRE.** More visualization can be found in Supp.

attack accuracy drops from 88.67% to 27.75% when $a_e = 0.4$ while natural accuracy increases 1.83%. In conclusion, *using RE-images during training significantly degrades MI attack while impact on natural accuracy is small.*

## 3.2 Feature space analysis of Random Erasing's defense effectiveness

In this section, we present a novel observation that explains RE's defense effectiveness. We observe **Property P1: Model trained with RE-private images following our MIDRE leads to a discrepancy between the features of MI-reconstructed images and that of private images**, resulting in degrading of attack accuracy.

The following analysis explains why MIDRE has **Property P1**. We use the following notation: $f_{train}$, $f_{priv}$ , $f_{RE}$ , and $f_{recon}$ represent the features of training images, private images , RE-private images , and MI-reconstructed images , respectively. To extract these features, we first train the target model without any defense (NoDef) and another target model with our MIDRE. Then, we pass images into these models to obtain the penultimate layer activations. Specifically, we input private images into the models to obtain $f_{priv}$. Next, we apply RE to private images, pass these RE-private images into the models to obtain $f_{RE}$. We also perform MI attacks to obtain reconstructed images from NoDef model (resp. MIDRE model), and then feed them into the NoDef model (resp. MIDRE model) to obtain $f_{recon}$. We use the same experimental setting as in Sec. 3.3. Then, we visualize penultimate layer activations $f_{priv}, f_{RE}, f_{recon}$ by both NoDef and our MIDRE model. We use $a_e = 0.4$ to train MIDRE and to generate RE-private images. Additionally, we visualize the convex hull of these features.

**Features of MI-reconstructed images tend to match features of training data.** SOTA MI attacks aim to reconstruct images that maximize the likelihood under the target model (Eq. 1) in order to extract training data (which possess a high likelihood under the target model). Under attacks of high accuracy,

$f_{recon}$ tends to match the features of training data $f_{train}$ (Nguyen et al., 2023).

**Evidence.** In Fig. 3 (a), as the training data of NoDef is private images $f_{train}^{NoDef} = f_{priv}^{NoDef}$, we observe that in NoDef model, $f_{recon}^{NoDef}$ overlaps $f_{priv}^{NoDef}$, i.e. there is significant overlap between the pink and blue polygons. In Fig. 3 (b), the MIDRE model is trained with RE-private images $f_{train}^{MIDRE} = f_{RE}^{MIDRE}$, and as a result, pink polygon $f_{recon}^{MIDRE}$ and green polygon $f_{RE}^{MIDRE}$ overlap. This confirms **features of reconstructed images tend to match the features of training data**, i.e. $f_{priv}^{NoDef}$ in NoDef and $f_{RE}^{MIDRE}$ in MIDRE.

**Mismatch in feature space of MIDRE.** MIDRE is trained using RE-private images and is generalizable to images without RE as shown in (Zhong et al., 2020). Under MIDRE target model, $f_{RE}^{MIDRE}$ and $f_{priv}^{MIDRE}$ have partial overlaps, but they are not identical. Meanwhile, $f_{recon}^{MIDRE}$ tend to match with $f_{RE}^{MIDRE}$ (RE-private images are training data for MIDRE, and follows the discussion above). Therefore, $f_{recon}^{MIDRE}$ do not replicate $f_{priv}^{MIDRE}$, significantly degrading the MI attack.

**Evidence.** In Fig. 3 (b), $f_{RE}^{MIDRE}$ and $f_{priv}^{MIDRE}$ are partial overlap. Importantly, $f_{recon}^{MIDRE}$, which overlaps with $f_{RE}^{MIDRE}$ as explained above, only partially overlaps with $f_{priv}^{MIDRE}$, suggesting MI attacks fail to guide the reconstructed features to replicate private features. Consequently, **MIDRE introduces a discrepancy between MI-reconstructed and private images in feature space of the target model, degrading the attack accuracy.**

**Remark.** Note that the mismatch between $f_{RE}^{MIDRE}$ and $f_{priv}^{MIDRE}$ does not cause the reduction of model utility (see Fig. 4). This is because the private images remain distinct from other classes and distant from other classification regions, even when their representations are partially overlapped with RE-private images (the training data).

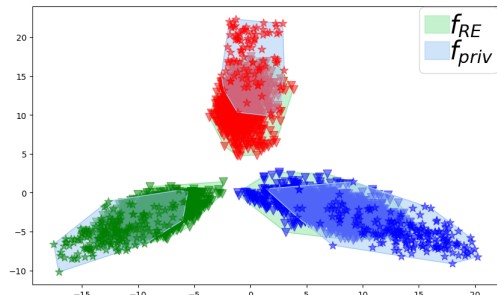

Figure 4: **MIDRE target model achieves high accuracy despite partial overlap of** $f_{RE}^{MIDRE}$ **and** $f_{priv}^{MIDRE}$. We visualize the penultimate layer activations of RE-private images and private images for three identities. While $f_{RE}^{MIDRE}$ and $f_{priv}^{MIDRE}$ do not completely overlap, the model can still classify private images with high accuracy. This is because the private images remain distinct from other classes and distant from other classification regions, even when their representations are partially shared with RE-private images (the training data). We remark that RE randomly erases different regions from the images in different iterations, preventing the model to learn shortcut features and forcing the model to learn intrinsic features and become more generalizable beyond training data. More visualization can be found in Supp.

### 3.3 Importance of partial erasure and random location for privacy-utility trade-off

In addition to two properties discussed in Sec. 3.2 which contribute to outstanding effectiveness of applying RE to degrade MI, we analyse in this section two properties of Random Erasing that are: **Property P2: Partial Erasure**, and **Property P3: Random Location**. To investigate the effect of each property, we conduct the experiment using the following setup: We use $T$ = ResNet-18 (Simonyan & Zisserman, 2014), $D_{priv}$ = Facecrub (Ng & Winkler, 2014), $D_{pub}$ = FFHQ (Karras et al., 2019), attack method = PPA (Struppek et al., 2022). To evaluate the effectiveness of **Partial Erasure** and **Random Location**, we conduct experiments on three schemes: **Entire Erasing (EE), Fixed Erasing (FE)**, and **Random Erasing (RE)**. These schemes are compared against a No Defense baseline, which is trained for 100 epochs without any defense. In the Entire Erasing (EE) scheme, we randomly reduce the number of erased samples per ID in each training epoch to simulate varying pixel concealment levels. Specifically, we train the model for 100 epochs, and for each epoch, we erase randomly 50%, 40%, 30%, 20%, 10%, and 0% of images per ID. For Fixed Erasing (FE), a fixed location within each image is erased throughout the entire training process. However, the erased location varied across different images. For Random Erasing (RE), the location of erased areas is randomly selected for each image and training iteration. We train the RE model for 100 epochs with different values of the erasure ratio, $a_e$ = 0.5, 0.4, 0.3, 0.2, 0.1 corresponding to 50%, 40%, 30 %, 20%, and 10% pixel concealment, respectively.

Table 1: We compare three different techniques for pixel concealment, to reduce the amount of private information presented to the model during training. The results show that simply reducing epochs as in "Entire Erasure" is insufficient for degrading attack performance. Meanwhile, RE improves model utility while degrading attack accuracy effectively.

| Concealment | | Partial Erasure | | | | Entire Erasing | |
| | | Random Erasing | | Fixed Erasing | | | |
| | $a_e$ | Acc ($\uparrow$) | AttAcc ($\downarrow$) | Acc ($\uparrow$) | AttAcc ($\downarrow$) | Acc ($\uparrow$) | AttAcc ($\downarrow$) |
|---|---|---|---|---|---|---|---|
| 0% | 0 | 97.69 | 87.12 | 97.69 | 87.12 | 97.69 | 87.12 |
| 10% | 0.1 | **97.91** | **79.76** | 96.13 | 85.26 | 97.33 | 87.83 |
| 20% | 0.2 | **97.64** | **66.32** | 96.79 | 69.81 | 97.52 | 87.03 |
| 30% | 0.3 | 97.14 | **46.30** | 96.13 | 50.71 | **97.53** | 89.15 |
| 40% | 0.4 | 96.05 | **27.75** | 93.10 | 28.49 | **97.30** | 89.03 |
| 50% | 0.5 | 93.77 | 15.98 | 86.69 | **14.86** | **97.21** | 87.59 |

**Property P2** **brings the privacy effect to defend against MI attacks**. By erasing portions of training images, it reduces the amount of private information exposed to the model during training. By erasing more information, we can effectively degrade the accuracy of privacy attacks. Importantly, partial erasures prevent the model from seeing **entire images**. Consequently, RE-images provide less information about the entirety of the face, such as inter-pupillary distance, relative distances between the eyes, nose, and mouth, the position of the cheekbones, etc. Note that, features need to be presented to the model many times during training in order for the model to learn the features, as suggested by the common practice of using multiple epochs to train a robust model. **Property P2** reduces the frequency of presenting the features to the model during training. Such reduced frequency using partial erasure makes it more difficult for the model to memorize the identity features.

**Evidence.** In Tab. 1, in terms of degrading the attack, partial erase (fixed or random) is more effective than entire erase (reduce numbers of sample per ID) although the percentages of pixel concealment are the same. Specifically, EE (reduce 50% images per ID) is significantly more vulnerable to attacks than RE and FE (50% image areas are erased, trained in 100 epochs), suffering approximately 71% higher in attack accuracy.

**Property P3** **recovers the model utility**. While information reduction can improve privacy, it may also negatively impact model utility if too much information is erased. Fixing the erasing location for an image means some identity feature of this image will never be presented to the model, model may not have adequate information to learn effectively. RE avoids this issue. As the location of erased area is changed in each training iteration, RE improves the diversity of the training data and ensures that the model still observes a significant portion of the image.

**Evidence.** In Tab. 1, RE improves the model accuracy while maintains the same attack accuracy as FE in different erased portion ratio $a_e$. For instance, RE has higher model accuracy than FE by 7.08% with $a_e$ = 0.5. With $a_e$ = 0.3 and 0.4, RE has higher accuracy and lower attack accuracy than EE model, showing that privacy effect of RE.

## 4 Experiments

### 4.1 Experimental Setting

To demonstrate the generalisation of our proposed MI defense, we carry out multiple experiments using different SOTA MI attacks on various architectures. In addition, we also use different setups for public and private data. The summary of all experiment setups is shown in Tab. 2. In total, we conducted 37 experiment setups to demonstrate the effectiveness of our proposed defense MIDRE.

**Dataset**: We follow the same setups as SOTA attacks (Zhang et al., 2020; Nguyen et al., 2023; Struppek et al., 2022) and defense (Peng et al., 2022; Struppek et al., 2024; Ho et al., 2024) to conduct the experiments on four datasets including: CelebA (Liu et al., 2015), FaceScrub (Ng & Winkler, 2014), VGGFace2 (Cao

Table 2: In total, we conduct 37 experiment setups to demonstrate the effectiveness of MIDRE.

| Attack | $T$ | $\mathcal{D}_{priv}$ | $\mathcal{D}_{pub}$ | Resolution |
|---|---|---|---|---|
| GMI (Zhang et al., 2020) KedMI (Chen et al., 2021) LOMMA (Nguyen et al., 2023) PLGMI (Yuan et al., 2023) RLBMI (Han et al., 2023) BREPMI (Kahla et al., 2022) | VGG16 (Simonyan & Zisserman, 2014) IR152 (He et al., 2016) FaceNet64 (Cheng et al., 2017) | CelebA | CelebA/FFHQ | 64×64 |
| PPA (Struppek et al., 2022) | ResNet18 (He et al., 2016) ResNet101 (He et al., 2016) ResNet152 (He et al., 2016) DenseNet121 (Huang et al., 2017) DenseNet169 (Huang et al., 2017) MaxVIT (Tu et al., 2022) | Facescrub | FFHQ | 224×224 |
| | ResneSt101 | Stanford Dogs | AFHQ Dogs | |
| MIRROR (An et al., 2022) | Inception-V1 (Inc) ResNet50 (He et al., 2016) | VGGFace2 | FFHQ | 160×160 224×224 |
| IF-GMI (Qiu et al., 2024) | ResNet18 ResNet152 | Facescrub | FFHQ | 224×224 |

et al., 2018), and Stanford Dogs (Dataset, 2011). We use FFHQ (Karras et al., 2019) and AFHQ Dogs (Choi et al., 2020) for the public dataset. We strictly follow (Zhang et al., 2020; Nguyen et al., 2023; Struppek et al., 2022; An et al., 2022; Peng et al., 2022; Struppek et al., 2024; Ho et al., 2024; Koh et al., 2024) to divide the datasets into public and private set. See Supp for the details of datasets.

**Model Inversion Attacks.** To evaluate the effectiveness of our proposed defense MIDRE, we employ a comprehensive suite of state-of-the-art MI attacks. This includes various attack categories: white-box and label-only, one type of black-box attack. To assess robustness at high resolutions, we employ PPA (Struppek et al., 2022) and IF-GMI (Qiu et al., 2024) against attacks targeting 224×224 pixels and MIRROR (An et al., 2022) against attacks targeting 224×224 and 160×160 pixels. For low resolution 64×64 pixels, we leverage four SOTA white-box attacks: GMI (Zhang et al., 2020), KedMI (Chen et al., 2021), PLG-MI (Yuan et al., 2023), and LOMMA (Nguyen et al., 2023) (including LOMMA+GMI and LOMMA+KedMI). Additionally, we incorporate RLBMI (Han et al., 2023) for black-box attack and BREPMI (Kahla et al., 2022) for label-only attack. We strictly replicate the experimental setups in (Zhang et al., 2020; Chen et al., 2021; Yuan et al., 2023; Nguyen et al., 2023; Struppek et al., 2022; Peng et al., 2022; Han et al., 2023; An et al., 2022) to ensure a fair comparison between NoDef (the baseline model without defense), existing state-of-the-art defenses, and our proposed method, MIDRE.

**Target Models.** We follow other MI research (Zhang et al., 2020; Nguyen et al., 2023; Struppek et al., 2022; Peng et al., 2022) to train defense models. We use 11 architectures for the target model to assess its resistance to MI attacks using various experimental configurations. The details are summarized in Tab. 2. We train target models with the same hyper-parameter ($a_h$) for all low-resolution data set-ups. In addition, for high-resolution data, we use two value for hyper-parameter $a_h = 0.4$ and $a_h = 0.8$ across all setups. This allows us to demonstrate MIDRE's effectiveness in achieving the optimal trade-off between utility and privacy with consistent hyper-parameter.

**Comparison Methods.** We compare the performance of our model against no defending method (NoDef) and five defense methods, including NLS (Negative Label Smoothing)(Struppek et al., 2024), TL-DMI (Ho et al., 2024), MI-RAD (MI-resilient architecture designs) (Koh et al., 2024), BiDO (Peng et al., 2022), and MID (Wang et al., 2021). As for MI-RAD (Koh et al., 2024), we compare our results to Removal of Last Stage Skip-Connection (RoLSS), Skip-Connection Scaling Factor (SSF), Two-Stage Training Scheme (TTS).

We establish a baseline (NoDef) by training the target model from scratch without incorporating any MI defense strategy. According to NLS, TL-DMI, MI-RAD, we follow their setup and evaluation to compare with MIDRE. We then carefully tune the hyper-parameters of each method to achieve optimal performance.

**Evaluation Metrics.** MI defenses typically involve a trade-off between the model's utility and its resistance

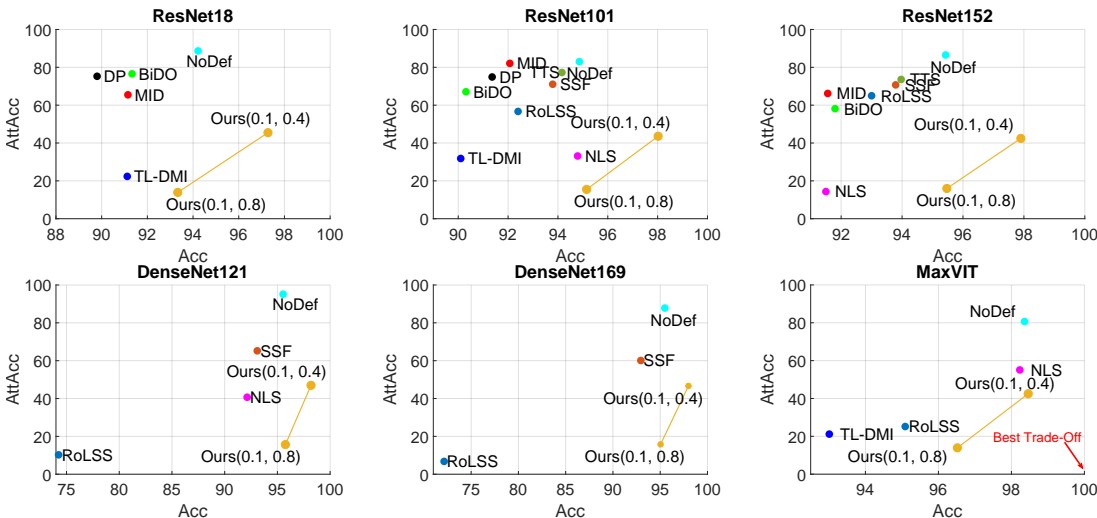

Figure 5: We evaluate PPA attack (Struppek et al., 2022) on our proposed method, NoDef, MID (Wang et al., 2021), BiDO (Peng et al., 2022), NLS (Struppek et al., 2024), and TL-DMI (Ho et al., 2024). Target models are trained on $\mathcal{D}_{priv}$ = Facescrub with 6 architectures. The results show that our method archives the best trade-of between utility and privacy among state-of-the-art defenses.

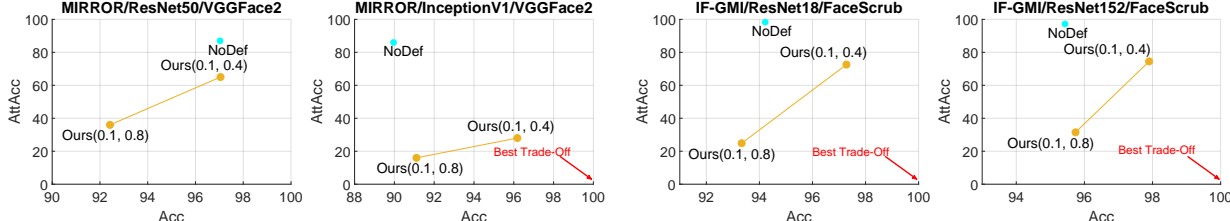

Figure 6: We evaluate MIRROR attack (An et al., 2022) on VGGFace2 dataset. The results show that our method archives the best trade-of between utility and privacy than NoDef model.

Figure 7: Results of IF-GMI (Qiu et al., 2024) attack on Facescrub dataset. Here, we use $T$ = ResNet18/ResNet152, $\mathcal{D}_{priv}$ = Facescrub, $\mathcal{D}_{pub}$ = FFHQ, image resolution = 224×224 images.

to model inversion attacks. In this section, we evaluate these defenses using two key metrics: Natural Accuracy (Acc ↑) to evaluate the model utility and Attack accuracy (AttAcc ↓) and to evaluate the model privacy. We further show other evaluation metrics, including K-Nearest Neighbor Distance, $\delta_{eval}$, $\delta_{face}$ (Struppek et al., 2022), complement these results with qualitative results and a user study in Supp.

## 4.2 Comparison against SOTA MI Defenses

We compare the model accuracy and attack accuracy of defense models in 6 architectures using attack method PPA (Struppek et al., 2022) in Fig. 5. All the target models are trained on Facescrub dataset. Interestingly, we are the first to observe that our defense models achieve higher natural accuracy but lower attack acuracy than no defense model for larger image sizes (224×224). With small masking areas (Ours(0.1,0.4)), our proposed method consistently achieves the lowest attack accuracy among all defense models while its natural accuracy is higher than NoDef, BiDO, MID, and DP models. For example, using ResNet101, our model reduces attack accuracy by 39.42% compared to NoDef while achieving the model accuracy is higher than NoDef model 3.16%. MaxVIT, a recent advanced architecture, has very high attack accuracy (80.66%). Our defense mechanism significantly enhances its robustness, lowering attack accuracy to 42.5% without compromising model performance. By increasing the masking areas (Ours(0.1,0.8)), they achieve a significant reduction in attack accuracy while maintaining high natural accuracy, outperforming other strong defense methods like NLS and TL-DMI. Specially, *our attack accuracies are below 20% for all architectures.* This

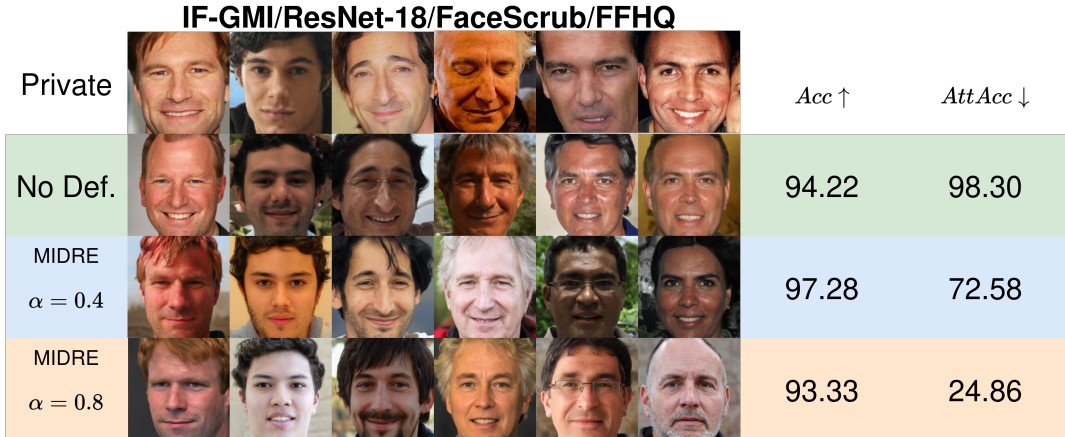

Figure 8: Reconstructed image obtained from IF-GMI attack with $T$ = ResNet-18, $\mathcal{D}_{priv}$ = Facescrub, $\mathcal{D}_{pub}$ = FFHQ. Overall, our reconstructed images have less similarity to private images compared to those from the no-defense model, suggesting the efficiency of our proposed defense MIDRE.

represents the best utility-privacy trade-off among all evaluated defense models, demonstrating our method's effectiveness in mitigating model inversion attacks.

We further show the effectiveness of our proposed method using two attacks MIRROR (An et al., 2022) and IF-GMI (Qiu et al., 2024). Regarding the MIRROR attack, we compare the results of MIDRE and the NoDef model using $\mathcal{D}_{priv}$ = VGGFace2 (see Fig. 6). Our defense reduces the attack accuracy by 22% and 70% without harming model utility, where the target model $T$ = ResNet50/InceptionV1. Results of IF-GMI attack are shown in Fig. 7. The results show that MIDRE reduces the attack accuracy by more than 22%.

For low resolution attacks, we evaluate against four MI attacks, including GMI (Zhang et al., 2020), KedMI (Chen et al., 2021), LOMMA (Nguyen et al., 2023) with two variances (LOMMA+GMI and LOMMA+KedMI), and PLGMI (Yuan et al., 2023) We follow the same setup and compare with NLS in Tab. 3. In addition, we also use NoDef baseline in NLS paper to compare and estimate $\Delta$ in Tab. 3.

Overall, our proposed method, MIDRE, achieves significant improvements in security for 64×64 setups compared to SOTA MI defenses. MIDRE achieves this by demonstrably reducing top-1 attack accuracy while maintaining natural accuracy on par

Table 3: We evaluate multiple SOTA MI attacks on 64×64 images, comparing their performance under NoDef, NLS, and our MIDRE defense. $T$ = VGG16, $D_{priv}$ = CelebA, $D_{pub}$ = CelebA.

| Attack | Defense | Acc ↑ | AttAcc ↓ | Δ ↑ |
|---|---|---|---|---|
| LOMMA + GMI | NoDef | 85.74 | 53.64 ± 4.64 | - |
| | NLS | 80.02 | 39.16 ± 4.25 | 2.53 |
| | **MIDRE** | 79.85 | **26.62 ± 1.93** | **4.59** |
| LOMMA + KedMI | NoDef | 85.74 | 72.96 ± 1.92 | - |
| | NLS | 80.02 | 63.60 ± 1.37 | 1.64 |
| | **MIDRE** | 79.85 | **41.82 ± 1.24** | **5.29** |
| PLGMI | NoDef | 85.74 | 71.00 ± 3.31 | - |
| | NLS | 80.02 | 72.00 ± 2.50 | -0.17 |
| | **MIDRE** | 79.85 | **66.60 ± 2.94** | **0.75** |

with other leading MI defenses. More results of other attacks can be found in Supp.

We show the comparison on qualitative results in Fig. 8. Images are reconstructed using IF-GMI atack (Qiu et al., 2024), $T$ = ResNet-18, $\mathcal{D}_{priv}$ = Facescrub, $\mathcal{D}_{pub}$ = FFHQ. Overall, our reconstructed images have less similarity to private images compared to those from the no-defense model, demonstrating the effectiveness of our defense. More reconstructed images are included in Supp and our project page.

The experiment results demonstrate that our defense model has a small impact on model utility while significantly enhancing the model's robustness against state-of-the-art MI attacks. Moreover, we are the first to report a substantial improvement in model utility among all existing defenses.

### 4.3 Ablation study on data augmentation type.

To evaluate the effectiveness of our method compared to other data augmentation-based defenses, we compare

Table 4: We report the PPA attack on images with resolution 224×224. $T$ = ResNet18, $D_{priv}$ = Facescrub, $D_{pub}$ = FFHQ to target models trained with different data augmentation. MIDRE consistently achieves the best balance between utility and privacy, significantly degrading MI attack performance while maintaining competitive natural accuracy.

| Attack | Defense | Acc ↑ | AttAcc ↓ |
|--------|---------|-------|----------|
| PPA | NoDef | 94.22 | 88.67 |
| | **MIDRE** | 97.28 | **48.16** |
| | CutMix | 98.74 | 67.12 |
| | Random Cropping | 92.24 | 74.22 |
| | Gaussian Blur | 97.57 | 87.12 |

Table 5: The combination MIDRE with existing defense BiDO and NLS. The combine models significantly reduces attack accuracy compared to individual defenses. We denote "OP" for $\Delta$ if the accuracy of the defense model outperforms that of the NoDef model.

| Setup | Defense | Acc (↑) | AttAcc (↓) | $\Delta$(↑) |
|-------|---------|---------|-----------|-------------|
| **Setup 1** | NoDef | 95.43 | 86.51 | - |
| | NLS | 91.50 | 13.94 | 18.47 |
| | MIDRE | 95.47 | 15.97 | OP |
| | MIDRE + NLS | 93.69 | 3.75 | 47.65 |
| **Setup 2** | NoDef | 86.90 | 81.80 ± 1.44 | - |
| | BiDO | 79.85 | 63.00 ± 2.08 | 2.67 |
| | MIDRE | 79.85 | 43.07 ± 1.99 | 5.49 |
| | MIDRE + BiDO | 82.15 | 39.00 ± 1.30 | 9.01 |

MIDRE with models trained using CutMix, random cropping, and Gaussian blur. These augmentations are commonly employed to enhance model generalization. Here, we use the setup: attack method = PPA, $T$ = ResNet18, $D_{priv}$ = Facescrub, $D_{pub}$ = FFHQ, image resolution = 224×224.

The results, summarized in Table 4, indicate that while alternative augmentations provide some level of protection, *MIDRE consistently achieves the best balance between utility and privacy, significantly degrading MI attack performance while maintaining competitive natural accuracy.*

### 4.4 Combination with Existing Defenses

Since MIDRE improves defense effectiveness from the training data perspective, our proposed method can be combined with other defense mechanism from the training objective perspective such as BiDO (Peng et al., 2022) and NLS (Struppek et al., 2024). We use 2 setups: **Setup 1**: $T$ = ResNet152, $\mathcal{D}_{priv}$ = Facescrub, $\mathcal{D}_{pub}$ = FFHQ, Attack method = PPA, image size = 224 × 224. **Setup 2**: $T$ = VGG16, $\mathcal{D}_{priv}/\mathcal{D}_{pub}$ = CelebA, Attack method = LOMMA + KedMI, image size = 64 × 64. We use $a_e = [0.1, 0.4]$ and $a_e = [0.1, 0.8]$ for setup 1 and setup 2 to train MIDRE and during attack.

The results (see Tab. 5) demonstrate the effectiveness of combining MIDRE with either NLS or BiDO for enhancing defense against MI attacks, as our MIDRE takes a data-centric perspective for defense, complementary to existing defenses. In both experimental setups, the combination models demonstrate a substantial reduction in attack accuracy compared to using MIDRE or the other defenses independently. In particular, in setup 1, the combination of MIDRE and Negative LS achieves a remarkable 3.75% attack accuracy when using the state-of-the-art PPA attack while preserving model utility. For Setup 2, MIDRE + BiDO improves the natural accuracy of the model by 2.3% while reducing the attack accuracy by 4.07% and 24% compared to MIDRE and BiDO, respectively. **This shows our effectiveness of combining MIDRE and existing defense for a better defense.** The combination ability of MIDRE supports that it examines a distinct aspect of the system by focusing on data input, setting it apart from other existing approaches to defend against model inversion attacks.

## 5 Conclusion

We propose a novel approach to MI Defense via Random Erasing (MIDRE). We conduct an analysis to demonstrate that RE possess two crucial properties to degrade MI attack while the impact on model utility is small. Furthermore, our features space analysis shows that model trained with RE-private images following MIDRE leads to a discrepancy between the features of MI-reconstructed images and that of private images, resulting in reducing of attack accuracy. Experiments validate that our approach achieves outstanding performance in balancing model privacy and utility. The results consistently demonstrate the superiority of our method over existing defenses across various MI attacks, network architectures, and attack configurations. The code and additional results can be found in the Supplementary section.

## Acknowledgements

This research is supported by the National Research Foundation, Singapore under its AI Singapore Programmes (AISG Award No.: AISG2-TC-2022-007); The Agency for Science, Technology and Research (A*STAR) under its MTC Programmatic Funds (Grant No. M23L7b0021). This research is supported by the National Research Foundation, Singapore and Infocomm Media Development Authority under its Trust Tech Funding Initiative. Any opinions, findings and conclusions or recommendations expressed in this material are those of the author(s) and do not reflect the views of National Research Foundation, Singapore and Infocomm Media Development Authority. This research is also part-funded by the European Union (Horizon Europe 2021-2027 Framework Programme Grant Agreement number 10107245. Views and opinions expressed are however those of the author(s) only and do not necessarily reflect those of the European Union. The European Union cannot be held responsible for them) and by the Engineering and Physical Sciences Research Council under grant number EP/X029174/1.

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
