# Random Erasing vs. Model Inversion: A Promising Defense or a False Hope?
# Supplementary Materials

**Viet-Hung Tran**[*]                                    *h.tran@qub.ac.uk*
*Singapore University of Technology and Design (SUTD); The Queen's University Belfast*

**Ngoc-Bao Nguyen**[*]                        *thibaongoc_nguyen@sutd.edu.sg*
*Singapore University of Technology and Design (SUTD)*

**Son T. Mai**                                    *thaison.mai@qub.ac.uk*
*The Queen's University Belfast*

**Hans Vandierendonck**                        *h.vandierendonck@qub.ac.uk*
*The Queen's University Belfast*

**Ira Assent**                                    *ira@cs.au.dk*
*Aarhus University*

**Alex Kot**                                    *eackot@ntu.edu.sg*
*Nanyang Technological University (NTU)*

**Ngai-Man Cheung**[†]                        *ngaiman_cheung@sutd.edu.sg*
*Singapore University of Technology and Design (SUTD)*

**Reviewed on OpenReview:** *https://openreview.net/forum?id=S9CwKnPHaO*

In this supplementary material, we provide additional experiments, analysis, ablation study, and details that are required to reproduce our results. These are not included in the main paper due to space limitations.

Our code and additional results are available at: `https://ngoc-nguyen-0.github.io/MIDRE/`

## Contents

---

[*] The first two authors contributed equally. [†] Corresponding author.

# A   Details on Experimental Setup

## A.1   Evaluation Method

To evaluate the attack, existing methods Zhang et al. (2020); Chen et al. (2021); Nguyen et al. (2023); Struppek et al. (2024); An et al. (2022) train an evaluation model $E$ that has a distinct architecture and is trained on the private dataset $\mathcal{D}_{priv}$. Similar to human inspection practices (Zhang et al., 2020), the evaluation model acts as a human proxy for assessing the quality of information leaked through MI attacks. We report the details of the evaluation models in the Tab. A.1. All the evaluation models are provided by Chen et al. (2021); Struppek et al. (2022); An et al. (2022).

Table A.1: Details of evaluation model $E$ in all the experimental setup. All the evaluation models are provided by Chen et al. (2021); Struppek et al. (2022); An et al. (2022).

| Attack | $T$ | $\mathcal{D}_{priv}$ | $\mathcal{D}_{pub}$ | Resolution | $E$ | $E$'s accuracy |
|---|---|---|---|---|---|---|
| GMI (Zhang et al., 2020)
KedMI (Chen et al., 2021)
LOMMA (Nguyen et al., 2023)
PLGMI (Yuan et al., 2023)
RLBMI (Han et al., 2023)
BREPMI (Kahla et al., 2022) | VGG16 (Simonyan & Zisserman, 2014)
IR152 (He et al., 2016)
FaceNet64 (Cheng et al., 2017) | CelebA | CelebA/FFHQ | 64×64 | FaceNet112 | 95.80 |
| PPA (Struppek et al., 2022) | ResNet18 (He et al., 2016)
ResNet101 (He et al., 2016)
ResNet152 (He et al., 2016)
DenseNet121 (Huang et al., 2017)
DenseNet169 (Huang et al., 2017)
MaxVIT (Tu et al., 2022) | Facescrub | FFHQ | 224×224 | Inception-V3 | 96.20% |
|  | ResneSt101 | Stanford Dogs | AFHQ Dogs |  | Inception-V3 | 79.79% |
| MIRROR (An et al., 2022) | Inception-V1 (Inc) | VGGFace2 | FFHQ | 160×160 | ResNet50 | 99.88% |
|  | ResNet50 (He et al., 2016) |  |  | 224×224 | Inception-V1 | 99.65% |
| IF-GMI (Qiu et al., 2024) | ResNet18
ResNet152 | Facescrub | FFHQ | 224×224 | Inception-V3 | 96.20% |

We evaluate defense methods using the following metrics:

- **Natural Accuracy (Acc ↑)**. This metric measures the accuracy of the defended model on a private test set, reflecting its performance on unseen data. Higher natural accuracy indicates better performance of the primary task.

- **Attack accuracy (AttAcc ↓) Zhang et al. (2020).** This metric measures the percentage of successful attacks, where success is defined as the ability to reconstruct private information from the model's outputs. Lower attack accuracy indicates a more robust defense. Following existing works (Zhang et al., 2020; Chen et al., 2021; Nguyen et al., 2023; Struppek et al., 2022), we utilize

a separate evaluation model $E$ to evaluate the inverted images. Higher attack accuracy on the evaluation model signifies a more effective attack, implying a weaker defense.

- **K-Nearest Neighbor Distance (KNN Dist ↑) Chen et al. (2021).** KNN distance measures the similarity between a reconstructed image of a specific identity and their private images. This is calculated using the $L_2$ norm in the feature space extracted from the penultimate layer of the evaluation model. In MI defense, a higher KNN Dist value indicates a greater degree of robustness against model inversion (MI) attacks and a lower quality of attacking samples on that model.

- $\delta_{eval}$ **and** $\delta_{face}$**Struppek et al. (2022).** We also use $\delta_{eval}$ and $\delta_{face}$ metrics from (Struppek et al., 2022) to quantify the quality of inverted images generated by PPA. These two metrics are the same concept as KNN Dist, but different in the model to produce a feature to calculate distance. $\delta_{face}$ use pretrained FaceNet (Schroff et al., 2015) as model to extract penultimate features, while $\delta_{eval}$ uses evaluation model for PPA attack.

- **Trade-off value ( $\Delta$ ↑) Ho et al. (2024).** To quantify the trade-off between model utility (natural accuracy) and attack performance (attack accuracy), we follow previous work and let NoDef model and defended model be $f_n$ and $f_d$ respectively, we compute $\Delta = \frac{AttAcc_{f_n} - AttAcc_{f_d}}{Acc_{f_n} - Acc_{f_d}}$. This metric calculates the ratio between the decrease in attack accuracy and the decrease in natural accuracy when applying an MI attack to a model without defenses (NoDef) and defense models. We remark that this metric is used when defense models have lower natural accuracy compared to the no-defense model. A higher $\Delta$ value indicates a more favorable trade-off.

## A.2 Dataset

We use three datasets including CelebA (Liu et al., 2015), Facescrub (Ng & Winkler, 2014), and Stanford Dogs (Dataset, 2011) as private training data and use two datasets including FFHQ (Karras et al., 2019) and AFHQ Dogs(Choi et al., 2020) as public dataset.

The CelebA dataset (Liu et al., 2015) is an extensive compilation of facial photographs, encompassing more than 200,000 images that represent 10,177 distinct persons. For MI task, we follow (Zhang et al., 2020; Chen et al., 2021; Nguyen et al., 2023) to divide CelebA into private dataset and public dataset. There is no overlap between private and public dataset. All the images are resized to 64×64 pixels.

Facescrub (Ng & Winkler, 2014) consists of a comprehensive collection of 106836 photographs showcasing 530 renowned male and female celebrities. Each individual is represented by an average of around 200 images, all possessing diversity of resolution. Following PPA (Struppek et al., 2022), we resize the image to 224×224 for training target models.

The FFHQ dataset comprises 70,000 PNG images of superior quality, each possessing a resolution of 1024x1024 pixels. FFHQ is used as a public dataset to train GANs using during attacks (Zhang et al., 2020; Chen et al., 2021; Struppek et al., 2022).

Stanford dogs (Dataset, 2011) contains more than 20,000 images encompassing 120 different dogs. AFHQ Dogs (Choi et al., 2020) contain around 5,000 dog images in high resolution. Follow (Struppek et al., 2022), we use Stanford dogs dataset as private dataset while AFHQ Dogs as the public dataset.

VGGFace2 (Cao et al., 2018) is a large-scale face recognition dataset designed for robust face recognition tasks. It consists of images that are automatically downloaded from Google Image Search, capturing a wide range of variations in factors such as pose, age, illumination, ethnicity, and profession. The diversity of the dataset makes it suitable for training and evaluating face recognition models across different conditions and demographics. It contains more than 3.3 milions images for 9000 identities.

## A.3 Train the Defense model using Random Erasing

We depict our method in Algorithm 1.

---

**Algorithm 1** Train the Defense model using Random Erasing

---

**Input:** Private training data $\mathcal{D}_{priv} = \{(x_i, y_i)\}_{i=1}^N$, model $T_\theta$, a maximum masking area portion $a_h$.
**Output:** The MIDRE-trained model $T_\theta$.
Initialize $t \leftarrow 0$
**while** $t < t_{RE}$ **do**
    Sample a mini-batch $\mathcal{D}_b$ with size $b$ from $\mathcal{D}_{priv}$
    $\mathcal{D}_{RE} = \{\}$
    **while** $(x, y)$ in $\mathcal{D}_b$ **do**
        $\tilde{x} = x$
        Randomly select $a_e$ within the range $[0.1, a_h]$
        $\tilde{x} = RE(x, a_e)$         ▷ *This is following the procedure discussed in Sec. 2.2*
        $\mathcal{D}_{mask} \leftarrow (\tilde{x}, y)$
    **end while**
    Compute $\mathcal{L}(\theta) = \frac{1}{b} \sum^{\mathcal{D}_{RE}} \ell(T_\theta(\tilde{x}_i), y_i)$
    Backward Propagation $\theta \leftarrow \theta - \alpha \nabla \mathcal{L}(\theta)$
**end while**

---

### A.4 Hyper-parameters for Model Inversion Attack

In the case of GMI(Zhang et al., 2020), KedMI(Chen et al., 2021), and PLG-MI(Yuan et al., 2023), BREPMI(Kahla et al., 2022), our approach is primarily based on the referenced publication outlining the corresponding attack. However, in certain specific scenarios, we adhere to the BiDO study due to its distinct model inversion attack configuration in comparison to the original paper. The LOMMA(Nguyen et al., 2023) approach involves adhering to the optimal configuration of the method, which encompasses three augmented model architectures: EfficientNetB0, EfficientNetB1, and EfficientNetB2. We adopt exactly the same experimental configuration, including the relevant hyper-parameters, as described in the referenced paper. We also follow PPA and MIRROR paper's configuration (Struppek et al., 2022; An et al., 2022) for our MI attack setups.

### A.5 Hyper-parameters for MIDRE

Our method only requires a hyper-parameter $a_h$, which is 0.4 for all low-resolution setups. According to high-resolution setups, we use $a_h = 0.4$ and $a_h = 0.8$ as two setups for our defense.

## B Additional Experimental Results

### B.1 Experiments on low resolution images

We evaluate our method against existing Model Inversion defenses. We follow the experiment setup in BiDO (Peng et al., 2022) and report the results on the standard setup using $T = $ VGG16 and $\mathcal{D}_{priv} = $ CelebA in Tab. B.2. We evaluate against six MI attacks, including GMI (Zhang et al., 2020), KedMI (Chen et al., 2021), LOMMA (Nguyen et al., 2023) with two variances (LOMMA+GMI and LOMMA+KedMI), PLGMI (Yuan et al., 2023), and a black-box attack, BREPMI (Kahla et al., 2022).

Overall, our proposed method, MIDRE, achieves significant improvements in security for 64×64 setups compared to SOTA MI defenses. MIDRE achieves this by demonstrably reducing top-1 attack accuracy while maintaining natural accuracy on par with other leading MI defenses. Specifically, compared to BiDO, MIDRE offers a substantial 43.74% decrease in top-1 attack accuracy with sacrificing only 7.05% in natural accuracy (measured using the KedMI attack method). Notably, while BiDO achieves similar natural accuracy to MIDRE, it suffers from a significantly higher top-1 attack accuracy (8.84% higher than MIDRE).

Table B.2: We report the MI attacks under multiple SOTA MI attacks on images with resolution 64×64. We compare the performance of these attacks against existing defenses including NoDef, BiDO, MID and our method. $T$ = VGG16, $D_{priv}$ = CelebA, $D_{pub}$ = CelebA.

| Attack | Defense | Acc ↑ | AttAcc ↓ | Δ ↑ | KNN Dist ↑ |
|---|---|---|---|---|---|
| LOMMA + GMI | NoDef | 86.90 | 74.53 ± 5.65 | - | 1312.93 |
| | MID | 79.16 | 54.53 ± 4.35 | 2.58 | 1348.21 |
| | BiDO | 79.85 | 53.73 ± 4.99 | 2.95 | 1422.75 |
| | **MIDRE** | 79.85 | **31.93 ± 5.10** | **6.04** | **1590.12** |
| LOMMA + KedMI | NoDef | 86.90 | 81.80 ± 1.44 | - | 1211.45 |
| | MID | 79.16 | 67.20 ± 1.59 | 1.89 | 1249.18 |
| | BiDO | 79.85 | 63.00 ± 2.08 | 2.67 | 1345.94 |
| | **MIDRE** | 79.85 | **43.07 ± 1.99** | **5.49** | **1503.89** |
| PLGMI | NoDef | 86.90 | 97.47 ± 1.68 | - | 1149.67 |
| | MID | 79.16 | 93.00 ± 1.90 | 0.58 | 1111.61 |
| | BiDO | 79.85 | 92.40 ± 1.74 | 0.72 | 1228.36 |
| | **MIDRE** | 79.85 | **66.60 ± 2.94** | **4.38** | **1475.76** |
| GMI | NoDef | 86.90 | 20.07 ± 5.46 | - | 1679.18 |
| | MID | 79.16 | 20.93 ± 3.12 | -0.11 | 1698.50 |
| | BiDO | 79.85 | 6.13 ± 2.98 | 1.98 | 1927.11 |
| | **MIDRE** | 79.85 | **3.20 ± 2.15** | **2.39** | **2020.49** |
| KedMI | NoDef | 86.90 | 78.47 ± 4.60 | - | 1289.46 |
| | MID | 79.16 | 53.33 ± 4.97 | 3.25 | 1364.02 |
| | BiDO | 79.85 | 43.53 ± 4.00 | 4.96 | 1494.35 |
| | **MIDRE** | 79.85 | **34.73 ± 4.15** | **6.20** | **1620.66** |
| BREPMI | NoDef | 86.90 | 57.40 ± 4.92 | - | 1376.94 |
| | MID | 79.16 | 39.20 ± 4.19 | 2.35 | 1458.61 |
| | BiDO | 79.85 | 37.40 ± 3.66 | 2.84 | 1500.45 |
| | **MIDRE** | 79.85 | **21.73 ± 2.99** | **5.06** | **1611.78** |

Table B.3: Results of IF-GMI(Qiu et al., 2024) attack on Facescrub dataset. Here, we use $T$ = ResNet18/ ResNet152, $\mathcal{D}_{priv}$ = Facescrub, $\mathcal{D}_{pub}$ = FFHQ, image resolution = 224×224 images, attack method = IF-GMI.

| Architecture | Defense | Acc ↑ | AttAcc ↓ | $\delta_{eval}$ ↑ | $\delta_{face}$ ↑ | FID ↑ |
|---|---|---|---|---|---|---|
| ResNet18 | NoDef | 94.22 | 98.30 | 110.04 | 0.647 | 40.239 |
| | **MIDRE (0.1, 0.4)** | 97.28 | 72.58 | 122.03 | 0.698 | 39.7238 |
| | **MIDRE (0.1, 0.8)** | 93.33 | **24.85** | **171.48** | **0.966** | **41.325** |
| ResNet152 | NoDef | 95.43 | 97.24 | 115.76 | 0.633 | **45.703** |
| | **MIDRE (0.1, 0.4)** | 97.90 | 74.50 | 133.22 | 0.662 | 40.669 |
| | **MIDRE (0.1, 0.8)** | 95.74 | **31.43** | **150.89** | **0.847** | 40.388 |

## B.2 Additional results

We further show the effectiveness of our proposed method on a wide range of target model architectures including IR152, FaceNet64, DenseNet-169, ResNeSt-101, and MaxVIT. The results are shown in Tab. B.4, B.5, and Tab.B.6 and B.7 (for comparison with TL-DMI) for 64×64 images and in Tab.B.9 and B.10 for 224×224 images.

The experiment results consistently demonstrate the effectiveness of our proposed method. For example, with $T$ = IR152, we sacrifice only 6.25% in natural accuracy, but the attack accuracies drop significantly, from 22.07% (PLGMI attack) to 40% (LOMMA + GMI attack). Similarly, when $T$ = FaceNet64, natural accuracy decreases by 6.94%, while the attack accuracies drop significantly, from 24.47% (PLGMI attack) to 45% (LOMMA attack). We report the results of additional setup in Tab. B.8. In particular, we use attack method = PLGMI, $T$ = VGG16/IR152/FaceNet64, $\mathcal{D}_{priv}$ = CelebA, $\mathcal{D}_{pub}$ = FFHQ. In addition to

measuring attack accuracy, we incorporate KNN distance to demonstrate the efficacy of our strategy across different evaluation scenarios. The specifics of KNN distance can be found in Sec. A.1.

For high resolution images, interestingly, with $\mathcal{D}_{priv}$ = Facescrub, we see a slight increase in natural accuracy (1.95%) along with a significant reduction in attack accuracy of around 40%. These results consistently show that MIDRE significantly reduces the impact of MI attacks. We report detailed results of PPA attack on our method compared to SOTA defense including MID, DP, BiDO, TL-DMI, NLS and RoLSS, SSF, TTS. the results are presented in Tab. B.9 and B.10. We also use $\delta_{eval}$ and $\delta_{face}$, with details in Sec. A.1 to evaluate quality of PPA inverted images.

Table B.4: Additional results on 64×64 images. We use $T$ = IR152. The target models are trained on $\mathcal{D}_{priv}$ = CelebA and $\mathcal{D}_{pub}$ = CelebA. The results conclusively show that our defense model is effective compared to NoDef models.

| Attack | Defense | Acc ↑ | AttAcc ↓ | KNN Dist ↑ |
|---|---|---|---|---|
| GMI | NoDef | 91.16 | 32.40 ± 4.88 | 1587.28 |
| | **MIDRE** | 84.91 | **7.87 ± 3.30** | **1888.47** |
| KedMI | NoDef | 91.16 | 78.93 ± 5.15 | 1262.44 |
| | **MIDRE** | 84.91 | **40.07 ± 4.99** | **1548.16** |
| LOMMA | NoDef | 91.16 | 80.93 ± 4.56 | 1253.03 |
| + GMI | **MIDRE** | 84.91 | **40.93 ± 6.11** | **1559.88** |
| LOMMA | NoDef | 91.16 | 90.87 ± 1.31 | 1116.90 |
| + KedMI | **MIDRE** | 84.91 | **52.13 ± 1.81** | **1481.70** |
| PLGMI | NoDef | 91.16 | 99.47 ± 0.93 | 1021.42 |
| | **MIDRE** | 84.91 | **77.40 ± 4.79** | **1470.46** |

Table B.5: Additional results on 64×64 images. We use $T$ = FaceNet64. The target models are trained on $\mathcal{D}_{priv}$ = CelebA and $\mathcal{D}_{pub}$ = CelebA. The results conclusively show that our defense model is effective compared to NoDef models.

| Attack | Defense | Acc ↑ | AttAcc ↓ | KNN Dist ↑ |
|---|---|---|---|---|
| GMI | NoDef | 88.50 | 29.60 ± 5.43 | 1607.86 |
| | **MIDRE** | 81.56 | **6.73 ± 3.42** | **1908.19** |
| KedMI | NoDef | 88.50 | 81.67 ± 2.63 | 1270.71 |
| | **MIDRE** | 81.56 | **36.33 ± 6.06** | **1545.93** |
| LOMMA | NoDef | 88.50 | 83.33 ± 3.40 | 1259.61 |
| + GMI | **MIDRE** | 81.56 | **37.60 ± 3.74** | **1570.85** |
| LOMMA | NoDef | 88.50 | 90.87 ± 1.31 | 1116.90 |
| + KedMI | **MIDRE** | 81.56 | **54.33 ± 1.44** | **1456.84** |
| PLGMI | NoDef | 88.50 | 99.47 ± 0.69 | 1091.51 |
| | **MIDRE** | 81.56 | **75.00 ± 4.30** | **1509.78** |

### B.3 User Study

In addition to attack accuracy measured by the evaluation model, we conduct a user study to further validate the attack's effectiveness. Overall, we conduct two setups for user study with low-resolution images and high-resolution images. Our interface for user study is illustrated in Fig. B.1.

In the low-resolution setup, we compare our proposed method and BiDO (Peng et al., 2022). For fair comparison, we use the same setup as BiDO: $T$ = VGG16, $\mathcal{D}_{priv}$ = CelebA, $\mathcal{D}_{pub}$ = CelebA and use the pre-trained model of BiDO to generate their images. We use the attack method PLG-MI to generate the inverted images and randomly select one image for each identity for overall 150 first identities. We upload it to Amazon Mechanical Turk and designate three individuals to vote on two of our model's and BiDO's reconstructed images, for a total of 450 votes. Participants were asked to select one of 4 options: BiDO, MIDRE, none, or both, for each image pair. Each pair was rated by three different users.

Table B.6: Additional results compared with TL-DMI on 64×64 images. We use $T$ = IR152. The target models are trained on $\mathcal{D}_{priv}$ = CelebA and $\mathcal{D}_{pub}$ = CelebA. The results conclusively show that our defense model is effective.

| Attack | Defense | Acc ↑ | AttAcc ↓ | Δ ↑ | KNN Dist↑ |
|---|---|---|---|---|---|
| | NoDef | 91.16 | 32.40 ± 4.88 | - | 1587.28 |
| GMI | **TL-DMI** | 86.70 | **8.93 ± 3.73** | 5.26 | **1819.00** |
| | **MIDRE** | 87.94 | 11.07 ± 3.60 | **6.62** | 1813.11 |
| | NoDef | 91.16 | 78.93 ± 5.15 | - | 1262.44 |
| KedMI | **TL-DMI** | 86.70 | 64.60 ± 4.93 | 3.21 | 1333.00 |
| | **MIDRE** | 87.94 | **46.67 ± 5.45** | **10.02** | **1455.88** |
| LOMMA | NoDef | 91.16 | 80.93 ± 4.56 | - | 1253.03 |
| + GMI | TL-DMI | 86.70 | **41.87 ± 5.37** | 8.76 | **1551.00** |
| | **MIDRE** | 87.94 | 49.40 ± 6.30 | **9.79** | 1497.50 |
| LOMMA | NoDef | 91.16 | 90.87 ± 1.31 | - | 1116.90 |
| + | TL-DMI | 86.70 | 77.73 ± 1.57 | 2.95 | 1305.00 |
| KedMI | **MIDRE** | 87.94 | **62.93 ± 2.15** | **8.68** | **1551.00** |

Table B.7: Additional results compared with TL-DMI on 64×64 images. We use $T$ = FaceNet64. The target models are trained on $\mathcal{D}_{priv}$ = CelebA and $\mathcal{D}_{pub}$ = CelebA. The results conclusively show that our defense model is effective.

| Attack | Defense | Acc ↑ | AttAcc ↓ | Δ ↑ | KNN Dist ↑ |
|---|---|---|---|---|---|
| | NoDef | 88.50 | 29.60 ± 5.43 | - | 1607.86 |
| GMI | **TL-DMI** | 83.41 | 15.73 ± 4.58 | 2.72 | 1752.00 |
| | **MIDRE** | 85.74 | **7.47 ± 2.59** | **8.02** | **1898.29** |
| | NoDef | 88.50 | 81.67 ± 2.63 | - | 1270.71 |
| KedMI | **TL-DMI** | 83.41 | 73.40 ± 4.10 | 1.62 | 1265.00 |
| | **MIDRE** | 85.74 | **42.93 ± 5.22** | **14.04** | **1512.52** |
| LOMMA | NoDef | 88.50 | 83.33 ± 3.40 | - | 1259.61 |
| + GMI | TL-DMI | 83.41 | 43.67 ± 5.60 | 7.79 | **1616.00** |
| | **MIDRE** | 85.74 | **43.33 ± 6.02** | **14.49** | 1550.77 |
| LOMMA | NoDef | 88.50 | 90.87 ± 1.31 | - | 1116.90 |
| + | TL-DMI | 83.41 | 79.60 ± 1.78 | 2.21 | **1345.00** |
| KedMI | **MIDRE** | 85.74 | **58.07 +/- 1.78** | **11.88** | 1386.67 |

Table B.8: We report the PLGMI attacks on images with resolution 64×64. We compare to NoDef and BiDO methods. $T$ = VGG16, IR152 and FaceNet64, $D_{pub}$ = FFHQ. We remark that BiDO only releases their implementation and pretrained model in the setup of $T$ = VGG16.

| Architecture | Defense | Acc ↑ | AttAcc ↓ | Δ ↑ | KNN Dist ↑ |
|---|---|---|---|---|---|
| | NoDef | 86.90 | 81.80 ± 2.74 | - | 1323.27 |
| VGG16 | BiDO | 79.85 | 60.93 ± 3.99 | 2.96 | 1440.16 |
| | **MIDRE** | 79.85 | **36.07 ± 4.76** | **6.49** | **1654.41** |
| IR152 | NoDef | 91.16 | 96.60 ± 2.11 | - | 1187.37 |
| | **MIDRE** | 84.91 | **54.02 ± 4.86** | **6.81** | **1579.28** |
| FaceNet64 | NoDef | 88.50 | 95.00 ± 2.56 | - | 1250.90 |
| | **MIDRE** | 81.56 | **51.60 ± 3.61** | **6.25** | **1501.85** |

In the high-resolution setup, we compare MIDRE and TL-DMI (Ho et al., 2024), which is a state-of-the-art MI defense. We use the setup: $T$ = ResNet101, $\mathcal{D}_{priv}$ = Facescrub, $\mathcal{D}_{pub}$ = FFHQ, attack method = PPA. For every defense, we create inverted images for each of the 530 classes, then select one image for each class.

Table B.9: We report the PPA MI attacks on images with resolution 224×224. We compare the performance of these attacks against existing defenses including NoDef, MID, DP, BiDO NLS, TLDMI, and MI-RAD variances. $D_{priv}$ = Facescrub $D_{pub}$ = FFHQ, Arhchitecture is Resnet18, ResNet152 and ResNet101. We denote "NA" for $\delta_{face}$ and $\delta_{eval}$ if these numbers are not available in the official paper Ho et al. (2024); Koh et al. (2024); Struppek et al. (2024). We denote "OP" for $\Delta$ if the accuracy of the defense model outperforms that of the NoDef model.

| Architecture | Defense | Acc ↑ | AttAcc ↓ | $\delta_{eval}$ ↑ | $\delta_{face}$ ↑ | $\Delta$ ↑ |
|---|---|---|---|---|---|---|
| ResNet18 | NoDef | 94.22 | 88.67 | 123.85 | 0.74 | - |
| | MID | 91.15 | 65.47 | 137.75 | 0.87 | 7.56 |
| | DP | 89.80 | 75.26 | 130.41 | 0.82 | 3.03 |
| | BiDO | 91.33 | 76.56 | 127.86 | 0.75 | 4.54 |
| | TL-DMI | 91.12 | 22.36 | NA | NA | 21.39 |
| | MIDRE(0.1, 0.4) | **97.28** | 48.16 | 131.72 | 0.80 | OP |
| | MIDRE(0.1,0.8) | 93.33 | **13.89** | **154.79** | **0.97** | **84.02** |
| ResNet152 | NoDef | 95.43 | 86.51 | 113.03 | 0.73 | - |
| | MID | 91.56 | 66.18 | 137.18 | 0.86 | 5.25 |
| | BiDO | 91.80 | 58.14 | 147.28 | 0.87 | 7.82 |
| | NLS | 91.50 | **14.34** | NA | **1.23** | 18.36 |
| | RoLSS | 93.00 | 64.98 | NA | NA | 8.86 |
| | SSF | 93.79 | 70.71 | NA | NA | 9.63 |
| | TTS | 93.97 | 73.59 | NA | NA | 8.85 |
| | MIDRE(0.1,0.4) | **97.90** | 42.44 | 139.66 | 0.82 | OP |
| | MIDRE(0.1,0.8) | 95.47 | **15.97** | **155.61** | 0.95 | OP |
| ResNet101 | NoDef | 94.86 | 83.00 | 128.60 | 0.76 | - |
| | MID | 92.70 | 82.08 | 122.96 | 0.76 | 0.43 |
| | DP | 91.36 | 74.88 | 131.38 | 0.82 | 2.32 |
| | BiDO | 90.31 | 67.07 | 139.15 | 0.84 | 3.50 |
| | TL-DMI | 90.10 | 31.82 | NA | NA | 10.75 |
| | NLS(-0.05) | 94.79 | 33.14 | 130.94 | 0.90 | 712.29 |
| | RoLSS | 92.40 | 58.68 | NA | NA | 9.89 |
| | SSF | 93.79 | 71.06 | NA | NA | 11.16 |
| | TTS | 94.16 | 77.26 | NA | NA | 8.20 |
| | MIDRE(0.1,0.4) | **98.02** | 43.58 | 139.01 | 0.81 | OP |
| | MIDRE(0.1,0.8) | 95.15 | **15.47** | **155.80** | **0.96** | OP |

Finally, we upload them to Amazon Mechanical Turk and follow the same procedure as low-resolution images setup.

**Comparing BiDO and our proposed MIDRE:** According to the results, 221 users voted in favour of BiDO, 108 in favour of our approach, 119 in favour of neither, and 2 in favour of both. It suggests that the reconstructed image quality from our model is not as good as the reconstructed image quality from BiDO, therefore **our proposed defense is more effective**. Our results are presented in Tab. B.11.

**Comparing SOTA TL-DMI and our proposed MIDRE:** According to the results in Tab. B.12, 509 users chose images inverted from our model, while 537 users voted in favor of TL-DMI. This suggests that the inverted images from our models are of lower quality than those from TL-DMI. In addition, there are 522 people voted for none of the two images is similar with the original image, meanwhile only 22 users chose that both images are similar to the real image.

According to the final results of both settings, MIDRE is a better defense mechanism against MI than SOTA BiDO and TL-DMI, which is in line with the findings of other evaluation metrics.

Table B.10: We report the PPA MI attacks on images with resolution 224×224. We compare the performance of these attacks against existing defenses including NoDef, MID, DP, BiDO NLS, TLDMI, and MI-RAD variances. $D_{priv}$ = Facescrub $D_{pub}$ = FFHQ, Arhchitecture is DenseNet169, DenseNet121, ResneSt101, and MaxVIT. We denote "NA" for $\delta_{face}$ and $\delta_{eval}$ if these numbers are not available in the official paper Ho et al. (2024); Koh et al. (2024); Struppek et al. (2024). We denote "OP" for $\Delta$ if the accuracy of the defense model outperforms that of the NoDef model.

| Architecture | Defense | Acc ↑ | AttAcc ↓ | $\delta_{eval}$ ↑ | $\delta_{face}$ ↑ | $\Delta$ ↑ |
|---|---|---|---|---|---|---|
| DenseNet169 | NoDef | 95.49 | 87.80 | 124.74 | 0.77 | - |
| | RoLSS | 72.14 | 6.77 | NA | NA | 3.47 |
| | SSF | 92.95 | 60.99 | NA | NA | 10.56 |
| | MIDRE(0.1,0.4) | **97.99** | 46.67 | 136.18 | 0.81 | NA |
| | MIDRE(0.1,0.8) | 95.04 | **15.78** | **154.96** | **0.95** | **160.04** |
| DenseNet121 | NoDef | 95.54 | 95.13 | 116.14 | 0.68 | - |
| | NLS(-0.05) | 92.13 | 40.69 | 179.53 | **0.97** | 15.96 |
| | RoLSS | 74.25 | 10.24 | NA | NA | 3.99 |
| | SSF | 93.09 | 65.21 | NA | NA | 12.21 |
| | MIDRE(0.1,0.4) | **98.19** | 46.98 | 134.86 | 0.81 | OP |
| | MIDRE (0.1,0.8) | 95.76 | 15.66 | **154.62** | 0.96 | OP |
| ResneSt101 | NoDef | 95.38 | 84.27 | 129.18 | 0.81 | - |
| | NLS(-0.05) | 88.82 | 13.23 | **172.73** | **1.10** | 10.01 |
| | MIDRE(0.1,0.4) | **98.11** | 45.43 | 137.78 | 0.80 | NA |
| | MIDRE(0.1,0.8) | 95.09 | 15.54 | 156.44 | 0.96 | **237.00** |
| MaxVIT | NoDef | 98.36 | 80.66 | 110.69 | 0.69 | - |
| | TL-DMI | 93.01 | 21.17 | NA | NA | 10.59 |
| | NLS(-0.05) | 98.23 | 55.09 | 127.68 | 0.81 | 63.93 |
| | RoLSS | 95.09 | 25.17 | NA | NA | 15.68 |
| | MIDRE(0.1,0.4) | **98.46** | 42.50 | 133.61 | 0.81 | OP |
| | MIDRE(0.1,0.8) | 96.52 | 13.92 | **155.31** | **0.96** | 31.63 |

Table B.11: We report results for an user study that was performed with Amazon Mechanical Turk. Reconstructed samples of PLG-MI/VGG16/CelebA/CelebA with first 150 classes. The study asked users for inputs regarding the similarity between a private training image and the reconstructed image from BiDO trained model and our trained model. Less number of reconstructed images from our defensed model are selected by users, suggesting our defense is more effective.

| Defense | Num of samples selected by users as more similar to private data |
|---|---|
| BiDO | 221 |
| Ours | **108** |
| Both | 119 |
| None | 2 |

## B.4 Qualitative Results

We provide inversion results from the recent IF-GMI attack in Fig. B.2 ($T$ = ResNet-18) and Fig. B.3 ($T$ = ResNet-152) and $\mathcal{D}_{priv}$ = Facescrub, $\mathcal{D}_{pub}$ = FFHQ. These results further demonstrate the effectiveness of our proposed method.

Table B.12: We report results for an user study that was performed with Amazon Mechanical Turk. Reconstructed samples of PPA/ResNet101/FaceScrub/FFHQ with all 530 classes. The study asked users for inputs regarding the similarity between a private training image and the reconstructed image from TL-DMI trained model and our trained model. Less number of reconstructed images from our defensed model are selected by users, suggesting our defense is more effective.

| Defense | Num of samples selected by users as more similar to private data |
|---------|------------------------------------------------------------------|
| TL-DMI  | 537 |
| Ours    | **509** |
| Both    | 522 |
| None    | 22 |

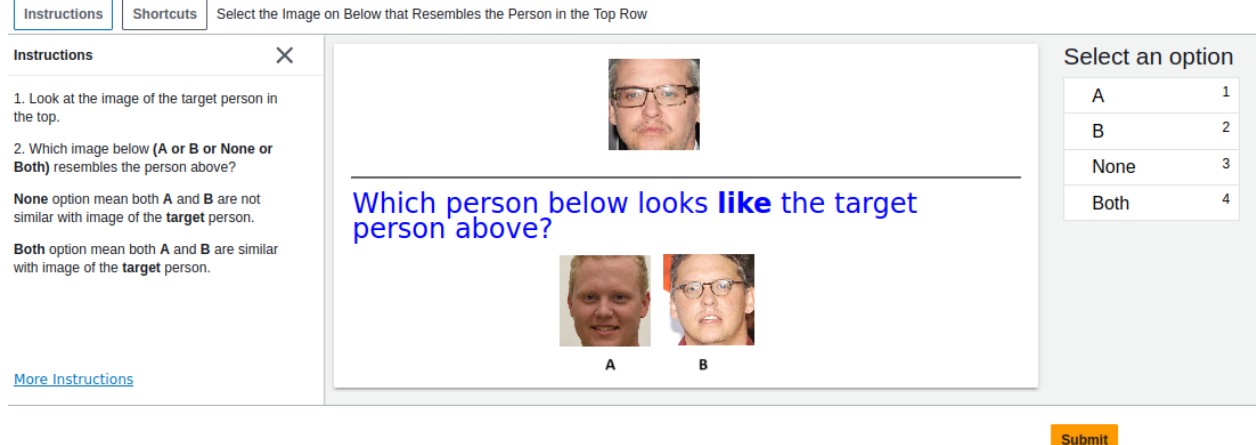

Figure B.1: Our Amazon Mechanical Turk (MTurk) interface for user study with model inversion attacking samples. Participants were asked to select one of 4 options: A, B, none, or both, for each image pair where A and B are the inverted images of our defense and other defense model. Each pair was rated by three different users.

## C    Additional analysis of privacy effect of MIDRE

### C.1    Feature space analysis of Random Erasing's defense effectiveness

In addition to the visualization of feature space analysis in Sec. 3.2 (main paper), we provide more visualization in other setup: $T$ = ResNet-152 (Simonyan & Zisserman, 2014), $D_{priv}$ = Facecrub (Ng & Winkler, 2014), $D_{pub}$ = FFHQ (Karras et al., 2019), attack method = PPA (Struppek et al., 2022). We observe **Property P1: Model trained with RE-private images following our MIDRE leads to a discrepancy between the features of MI-reconstructed images and that of private images**, resulting in degrading of attack accuracy.

We use the following notation: $f_{train}$, $f_{priv}$, $f_{RE}$, and $f_{recon}$ represent the features of training images, private images, RE-private images, and MI-reconstructed images, respectively. To extract these features, we first train the target model without any defense (NoDef) and another target model with our MIDRE. Then, we pass images into these models to obtain the penultimate layer activations. Specifically, we input private images into the models to obtain $f_{priv}$. Next, we apply RE to private images, pass these RE-private images into the models to obtain $f_{RE}$. We also perform MI attacks to obtain reconstructed images from NoDef model (resp. MIDRE model), and then feed them into the NoDef model (resp. MIDRE model) to obtain $f_{recon}$. Then, we visualize penultimate layer activations $f_{priv}, f_{RE}, f_{recon}$ by both NoDef and our

**IF-GMI/ResNet-18/FaceScrub/FFHQ**

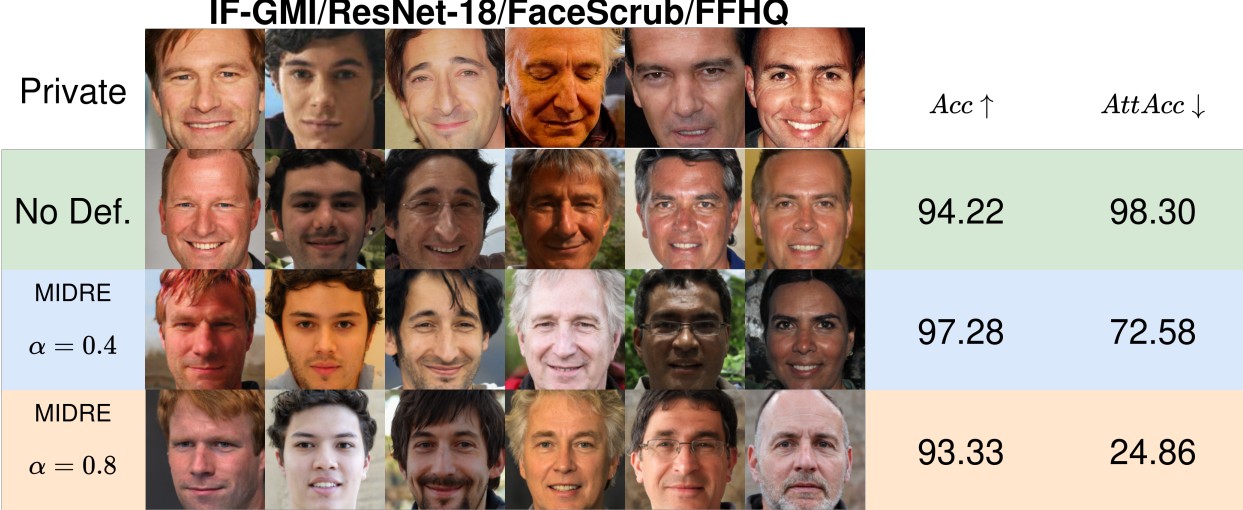

| | | $Acc \uparrow$ | $AttAcc \downarrow$ |
|---|---|---|---|
| Private | | | |
| No Def. | | 94.22 | 98.30 |
| MIDRE $\alpha = 0.4$ | | 97.28 | 72.58 |
| MIDRE $\alpha = 0.8$ | | 93.33 | 24.86 |

Figure B.2: Reconstructed image obtained from IF-GMI attack with $T = $ ResNet-18, $\mathcal{D}_{priv} = $ Facescrub, $\mathcal{D}_{pub} = $ FFHQ. The quality of the reconstructed image obtained from the attack on the model trained by MIDRE is comparatively worse when compared to that from NoDef method, suggesting the efficiency of our proposed defense MIDRE.

**IF-GMI/ResNet-152/FaceScrub/FFHQ**

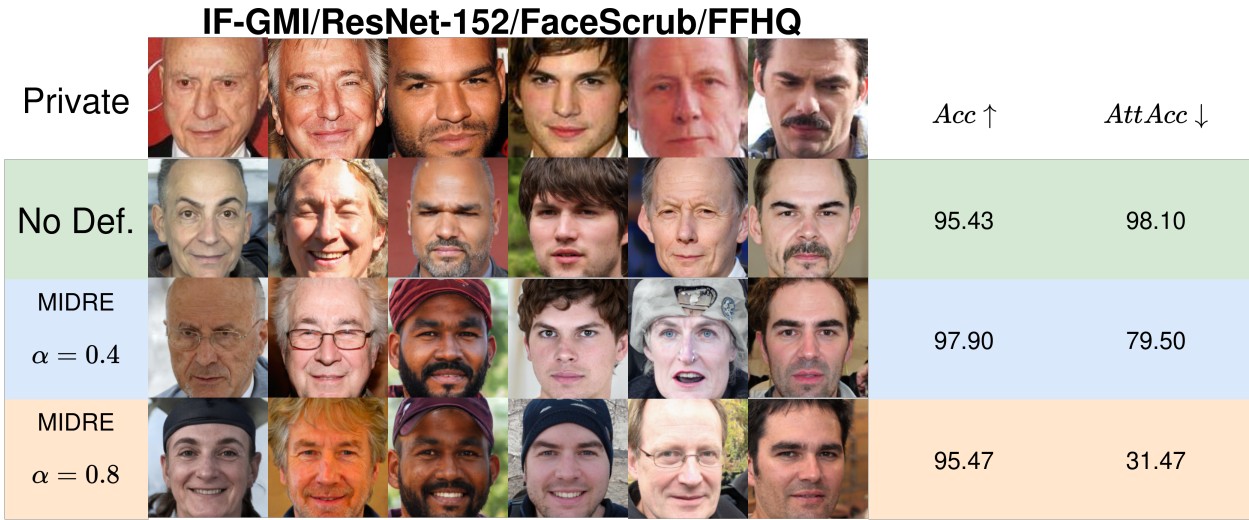

| | | $Acc \uparrow$ | $AttAcc \downarrow$ |
|---|---|---|---|
| Private | | | |
| No Def. | | 95.43 | 98.10 |
| MIDRE $\alpha = 0.4$ | | 97.90 | 79.50 |
| MIDRE $\alpha = 0.8$ | | 95.47 | 31.47 |

Figure B.3: Reconstructed image obtained from IF-GMI attack with $T = $ ResNet-152, $\mathcal{D}_{priv} = $ Facescrub, $\mathcal{D}_{pub} = $ FFHQ. The quality of the reconstructed image obtained from the attack on the model trained by MIDRE is comparatively worse when compared to that from NoDef method, suggesting the efficiency of our proposed defense MIDRE.

MIDRE model. We use $a_e = 0.4$ to train MIDRE and to generate RE-private images. Additionally, we visualize the convex hull of these features. For visualization, we employ PCA to reduce the dimension of the feature space.

The visualization in Fig. C.4 shows the same trend as in Sec. 3.2. Specially, we observe the mismatch in feature space of MIDRE. Under MIDRE target model, $f_{RE}^{MIDRE}$ and $f_{priv}^{MIDRE}$ have partial overlaps, but they are not identical. Meanwhile, $f_{recon}^{MIDRE}$ tend to match with $f_{RE}^{MIDRE}$ (RE-private images are training data for MIDRE, and follows the discussion above). Therefore, $f_{recon}^{MIDRE}$ do not replicate $f_{priv}^{MIDRE}$, significantly degrading the MI attack. Furthermore, Fig. C.5 shows that the mismatch between $f_{RE}^{MIDRE}$ and $f_{priv}^{MIDRE}$ does not cause the reduction of model utility. This is because the private images remain distinct from other classes and distant from other classification regions, even when their representations are partially overlapped with RE-private images (the training data).

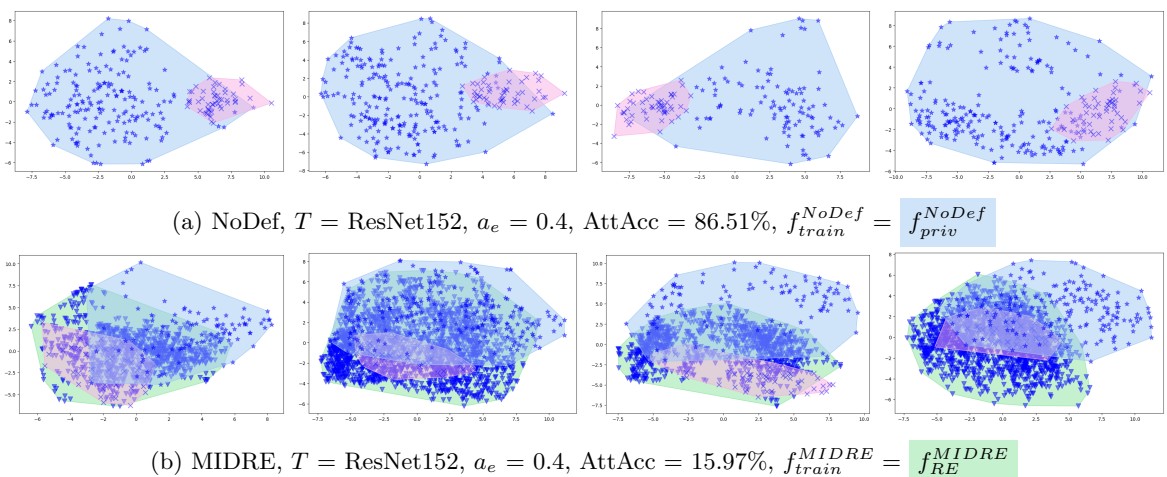

(a) NoDef, $T =$ ResNet152, $a_e = 0.4$, AttAcc = 86.51%, $f_{train}^{NoDef} = f_{priv}^{NoDef}$

(b) MIDRE, $T =$ ResNet152, $a_e = 0.4$, AttAcc = 15.97%, $f_{train}^{MIDRE} = f_{RE}^{MIDRE}$

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

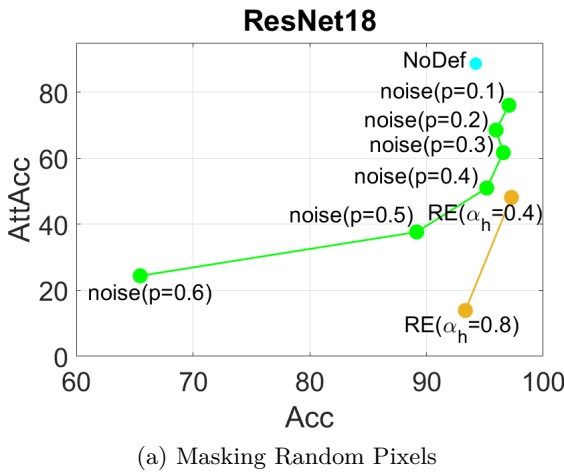
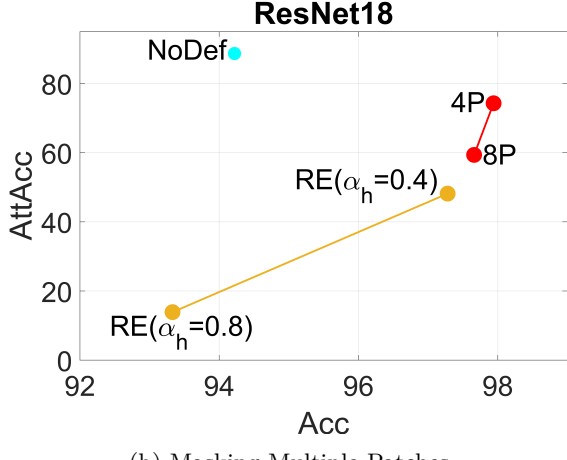

(a) Masking Random Pixels

(b) Masking Multiple Patches

Figure D.6: We compare our RE masking strategy with two alternative masking approaches: (a) *Masking Random Pixels*, and (b) *Masking Multiple Patches*. In (a), we randomly mask a proportion of pixels from 10% ($p = 0.1$) to 60% ($p = 0.6$). In (b), we randomly mask either 4 small patches, denoted as 4P ($\alpha_e \in [0.025, 0.1]$), or 8 small patches, denoted as 8P ($\alpha_e \in [0.0125, 0.1]$). We evaluate these strategies using the PPA attack method with $T = \text{ResNet18}$, $\mathcal{D}_{\text{priv}} = \text{FaceScrub}$, and $\mathcal{D}_{\text{pub}} = \text{FFHQ}$. The results demonstrate that our RE masking strategy achieves a better privacy-utility trade-off compared to both Masking Random Pixels and Masking Multiple Patches.

## D  Ablation Study

### D.1  Ablation study on alternative masking strategies

In this section, we conduct experiments using alternative masking strategies. In addition to the traditional random erasing method, we explore two additional approaches: (1) masking random pixels, and (2) masking multiple patches.

- **Masking random pixels:** Instead of masking a square region as in our proposed Random Erasing (RE) method, we apply masking at the pixel level. For example, we randomly mask 10% of the image pixels by replacing them with random values. In our experiments, we train the target model with varying levels of random pixel masking, ranging from 10% ($p = 0.1$) to 60% ($p = 0.6$).

- **Masking multiple patches:** Instead of masking a single large square region, we apply multiple smaller masks to the image. In our experiments, we randomly mask each training image with either 4 small patches (4P) or 8 small patches (8P). To ensure a fair comparison with MIDRE, we adjust the patch sizes accordingly. For 4P, we set $\alpha_e \in [0.025, 0.1]$, so that the total area of the four small patches is approximately equivalent to that of MIDRE with $\alpha_e \in [0.1, 0.4]$ (RE($\alpha_h = 0.4$)). Similarly, for 8P, we use $\alpha_e \in [0.0125, 0.1]$, making the total masked area comparable to MIDRE with $\alpha_e \in [0.1, 0.8]$ (RE($\alpha_h = 0.8$)).

We summarize the results of the two alternative masking strategies in Fig. D.6.

- **Masking Random Pixels:** We clearly observe that this method performs better than the baseline (NoDef) in terms of reducing attack accuracy. However, it is less effective than our proposed MIDRE in both lowering attack accuracy and preserving natural accuracy.

- **Masking Multiple Patches:** Although distributing the masking across multiple smaller regions provides some privacy benefits, masking a single large region—as done in our approach—still achieves a better utility-privacy trade-off.

## D.2 Ablation study on Masking Values.

In this section, we examine the effect of masking value to MIDRE performance. We select attack method = PLGMI (Yuan et al., 2023), $T$ = FaceNet64, $\mathcal{D}_{priv}$ = CelebA, $\mathcal{D}_{pub}$ = FFHQ. We set $a_e = (0.2, 0.2)$. Similar to (Zhong et al., 2020), we investigate four types of masking values: 0, 1, a random value, and the mean value. In case of random value, we randomly select it within a range (0,1). The mean value uses the ImageNet dataset's mean pixel values ([0.485, 0.456, 0.406]).

Tab. D.14 demonstrates that the mean value offers the best balance between robustness against MI attacks and maintaining natural image accuracy. Consequently, we adopt the Imagenet mean pixel values for masking in MIDRE.

Table D.14: The effect of different masking value. We use attack method = PLGMI (Yuan et al., 2023), $T$ = FaceNet64, $\mathcal{D}_{priv}$ = CelebA, $\mathcal{D}_{pub}$ = FFHQ. Overall, mean value achieves the best balance between robustness against MI attacks and maintaining natural image accuracy.

| Masking value | Acc ↑ | AttAcc ↓ | $\Delta$ ↑ | Ranking |
|---|---|---|---|---|
| NoDef | 88.50 | $95.00 \pm 2.56$ | - | - |
| 0 | 83.72 | $69.20 \pm 2.64$ | 5.40 | 3 |
| 1 | 83.68 | $70.00 \pm 3.18$ | 5.18 | 4 |
| random | 80.76 | $51.87 \pm 4.43$ | 5.57 | 2 |
| mean | 85.14 | $68.87 \pm 3.97$ | **7.78** | 1 |

## D.3 Ablation study on Area Ratio.

In MIDRE, the area ratio $a_e$ controls the portion of an image masked to prevent MI attacks. This experiment investigates the impact of different $a_e$ values on MIDRE's performance. In particular, $a_e$ is randomly selected within the range (0.1, $a_h$), guaranting that at least 10% of the image is always masked. We select three values for $a_h$: 0.3, 0.4, and 0.5. Similar to the previous ablation study, we employ attack method = PLGMI (Yuan et al., 2023), $T$ = FaceNet64, $\mathcal{D}_{priv}$ = CelebA, $\mathcal{D}_{pub}$ = FFHQ. The masking process uses the ImageNet mean pixel values.

The results in Tab. D.15 indicate that increasing $a_h$ strengthens MIDRE's defense against MI attacks, but this comes at the cost of reduced natural accuracy. To achieve a balance between robustness and natural accuracy, we opt $a_h = 0.4$ in MIDRE.

Table D.15: The effect of area ratio. We use attack method = PLGMI (Yuan et al., 2023), $T$ = FaceNet64, $\mathcal{D}_{priv}$ = CelebA, $\mathcal{D}_{pub}$ = FFHQ. To achieve a balance between robustness and natural accuracy, we opt $a_h$ = 0.4 in MIDRE.

| $a_h$ | Acc ↑ | AttAcc ↓ | $\Delta$ ↑ | Ranking |
|---|---|---|---|---|
| NoDef | 88.50 | $95.00 \pm 2.56$ | - | - |
| 0.3 | 83.55 | $65.07 \pm 4.02$ | 6.05 | 2 |
| 0.4 | 81.65 | $51.60 \pm 3.61$ | **6.34** | 1 |
| 0.5 | 78.50 | $45.40 \pm 3.85$ | 4.96 | 3 |

## D.4 Ablation study on Aspect Ratio.

We perform an ablation study on the aspect ratio of random erasing for model inversion defense. The results presented in Tab. D.16 demonstrate that the influence of aspect ratio on attack accuracy is not as significant as that of area ratio.

Table D.16: We report the LOMMA+KedMI attacks on images with resolution 64×64. $T$ = VGG16, $D_{priv}$ = CelebA, $D_{pub}$ = CelebA with different aspect ratios of RE in MIDRE. We also put NoDef result as a baseline.

| Attack | Defense | Acc ↑ | AttAcc ↓ | Δ ↑ | KNN Dist ↑ |
|---|---|---|---|---|---|
| | NoDef | 86.90 | 81.80 ± 1.44 | - | 1211.45 |
| | MIDRE | 79.85 | 43.07 ± 1.99 | 5.49 | 1503.89 |
| LOMMA+KedMI | MIDRE(aspect ratio = 0.5) | 81.32 | 49.13 ± 1.53 | 5.85 | 1424.40 |
| | MIDRE(aspect ratio = 2.0) | 81.65 | 51.87 ± 1.62 | 5.70 | 1440.00 |

## D.5   Adaptive attack

We perform adaptive attacks in which the attacker knows the portions of the masking area $a_e$ and uses it during inversion attacks. We use 2 setups: **Setup 1**: $T$ = ResNet152, $\mathcal{D}_{priv}$ = Facescrub, $\mathcal{D}_{pub}$ = FFHQ, Attack method = PPA, image size = 224 × 224. **Setup 2**: $T$ = VGG16, $\mathcal{D}_{priv}/\mathcal{D}_{pub}$ = CelebA, Attack method = LOMMA + KedMI, image size = 64 × 64. We use $a_e$ = [0.1,0.8] and $a_e$ = [0.1,0.4] for setup 1 and setup 2 to train MIDRE and during attack.

**Adaptive attacks fail to enhance attack performance in both two experimental setups** (See Tab. D.17). This may be due to the dynamic masking positions employed in each attack iteration, hindering the convergence of the inverted images. Overall, even when attackers are fully informed about RE and use this knowledge to design an adaptive MI attack, they still fail to achieve accurate inversion results.

We compare the loss curves of the adaptive and normal attacks in Fig. D.7. The results show that the dynamic masking positions in each iteration cause greater fluctuations in the adaptive attack loss compared to the normal attack. In addition, PPA already incorporates learning rate adjustments during inversion, which do not reduce the loss fluctuations.

Table D.17: We conduct the adaptive attacks where the attacker knows the masking area portions $a_e$ and uses them during inversion attacks. **Adaptive attacks (Adapt.Att) fail to enhance attack performance in both setups.**

| Setup | Attack | AttAcc |
|---|---|---|
| Setup 1 | MIDRE | 15.97 |
| | MIDRE (Adapt.Att) | 10.50 **(-5.47%)** |
| Setup 2 | MIDRE | 43.07 |
| | MIDRE (Adapt.Att) | 38.53 **(-4.54%)** |

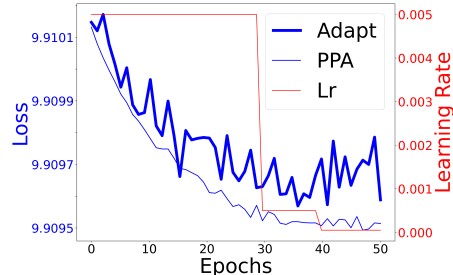

Figure D.7: PPA and PPA(Adapt) loss curves, with learning rate (Lr) adjustment.

## D.6   The effectiveness of substitute pixels generated by inpainting for MIDRE.

We incorporated an inpainting method (Telea, 2004) to replace masked values, following the experimental setup described earlier. Our results show that MIDRE (inpainting) modestly improves model accuracy while reducing the attack success rate by 4.34%, which is indicated in Tab. D.18. However, this approach incurs a higher computational cost compared to RE.

Table D.18: We report the LOMMA+KedMI attack on images with resolution 64×64. $T$ = VGG16, $D_{priv}$ = CelebA, $D_{pub}$ = CelebA to target models trained with RE with substitue pixel generate by inpainting.

| Attack | Defense | Acc ↑ | AttAcc ↓ | KNN Dist ↑ |
|---|---|---|---|---|
| | NoDef | 86.90 | 81.80 ± 1.44 | 1211.45 |
| LOMMA+KedMI | MIDRE | 79.85 | **43.07 ± 1.99** | 1503.89 |
| | **MIDRE (inpainting)** | 80.42 | **38.73 ± 1.27** | **1508.28** |

# E    Discussion

We propose a new defense against MI attacks using Random Erasing (RE) during training. RE reduces private information exposure while significantly lowering MI attack success, with small impact on model accuracy. Our method outperforms existing defenses across 34 experiment setups using 7 SOTA MI attacks, 11 model architectures, 6 datasets, and user study.

## E.1    Broader Impacts

Model inversion attacks, a rising privacy threat, have garnered significant attention recently. By studying defenses against these attacks, we can develop best practices for deploying AI models and build robust safeguards for applications, especially those that rely on sensitive training data. Research on model inversion aims to raise awareness of potential privacy vulnerabilities and strengthen the defense.

## E.2    Limitation

Firstly, we currently focus on enhancing the robustness of classification models against MI attacks. This is really important because these models are being used more and more in real-life situations where privacy and security are a major concern. In the future, we plan to expand our research scope to encompass MI attacks and defenses for a broader range of machine learning tasks.

Secondly, while our current experiments are comprehensive compared to prior works (Zhang et al., 2020; Chen et al., 2021; Nguyen et al., 2023; Kahla et al., 2022; Struppek et al., 2022; Ho et al., 2024; Koh et al., 2024) which mainly focus on image data, real-world applications often involve diverse private/sensitive training data. Addressing these real-world data complexities through a comprehensive approach will be essential for building robust and trustworthy machine learning systems across various domains.

# F    Experiments Compute Resources

In order to carry out our experiments, we utilise a workstation equipped with the Ubuntu operating system, an AMD Ryzen CPU, and 4 NVIDIA RTX A5000 GPUs. Furthermore, we utilise a secondary workstation equipped with the Ubuntu operating system, an AMD Ryzen CPU, and two NVIDIA RTX A6000 GPUs.

# G    Related Work

## G.1    Model Inversion Attacks

The GMI (Zhang et al., 2020) is a pioneering approach in model inversion to leverages publicly available data and employs a generative model GAN to invert private datasets. This methodology effectively mitigates the generation of unrealistic data instances. KedMI (Chen et al., 2021) can be considered an enhanced iteration of the GMI model, as it incorporates the transmission of knowledge to the discriminator through the utilization of soft labels. PLGMI (Yuan et al., 2023) is the current leading model inversion method in the field. It leverages pseudo labels derived from public data and the target model. LOMMA (Nguyen et al., 2023) employs an augmented model in order to reduce the model inversion overfitting. The augmented model is trained to distill knowledge from a target model by utilizing public data. During attack, the attackers generate images in order to minimize the identity loss in both the target model and the augmented model. However, it should be noted that the aforementioned four approaches are specifically designed for target models that have been trained on low-resolution data, specifically 64x64 for the CelebA private dataset. Recently, PPA (Struppek et al., 2022), MIRROR (An et al., 2022), and DMMIA (Qi et al., 2023), IF-GMI(Qiu et al., 2024) perform the attack on high resolution images. In addition, Kahla et. al. (Kahla et al., 2022) introduced the BREPMI attack as a form of label-only model inversion attack, where the assault is based on the predicted labels of the target model. Another work is RLBMI (Han et al., 2023), which utilizes a reinforcement learning approach to target a model in a black box scenario.

### G.2 Model Inversion Defenses

To defend against MI attacks, differential privacy (DP) (Dwork, 2006; 2008) has been studied in earlier works. Studies in (Dwork, 2006; 2008) have shown that current DP mechanisms do not mitigate MI attacks while maintaining desirable model utility at the same time. More recently, regularizations have been proposed for MI defenses (Wang et al., 2021; Peng et al., 2022; Struppek et al., 2024). (Wang et al., 2021) propose regularization loss to the training objective to limit the dependency between the model inputs and outputs. In BiDO (Peng et al., 2022), they propose regularization to limit the the dependency between the model inputs and latent representations. However, these regularizations conflict with the training loss and harm model utility considerably. To restore the model utility partially, (Peng et al., 2022) propose to add another regularization loss to maximize the dependency between latent representations and the outputs. However, searching for hyperparameters for two regularizations in BiDO requires computationally-expensive. Recently, (Ye et al., 2022) introduced a new approach that utilises differential privacy to protect against model inversion. (Gong et al., 2023) proposed a novel Generative Adversarial Network (GAN)-based approach to counter model inversion attacks. In this paper, we do not conduct experiments to compare to these methods as the code is not available. (Struppek et al., 2024) study the effect of label smoothing regularization on model privacy leakage. Their findings demonstrate that positive label smoothing factors can amplify privacy leakage, whereas negative label smoothing factors mitigate privacy concerns at the cost of a substantial decrease in model utility, resulting in a more favorable utility-privacy trade-off. Recently, (Ho et al., 2024) introduce a novel approach to defending against model inversion attacks by focusing on the model training process. Their proposed Transfer Learning-based Defense against Model Inversion (TL-DMI) aims to restrict the number of layers that encode sensitive information from the private training dataset into the model. As restricting the number of model parameters that encode private information can potentially impact the model's performance. (Koh et al., 2024) study the impact of DNN architecture designs, particularly skip connections, on model inversion attacks. They found that removing skip connections in the last layers can enhance model inversion robustness. However, this approach necessitates searching for optimal skip connection removal and scaling factor combinations, which can be computationally intensive. Both TL-DMI and MI-RAD experiences difficulty in achieving competitive balance between utility and privacy. We show comparison of several defense approaches with our MIDRE in Fig. 1 (main paper).