# OpenReview forum: "Random Erasing vs. Model Inversion: A Promising Defense or a False Hope?"
_TMLR — Accepted by TMLR_

### Review · Reviewer_njFH · 2025-05-16

**Summary Of Contributions:**

This paper proposes Model Inversion Defense via Random Erasing (MIDRE), a method to mitigate model inversion (MI) attacks by training models with randomly erased images. The approach leverages Random Erasing (RE), a data augmentation technique, to degrade MI attack accuracy while preserving model utility. The authors analyze RE’s effectiveness through feature-space visualization, demonstrating that RE introduces discrepancies between MI-reconstructed and private image features. Extensive experiments across 37 setups, including diverse datasets, architectures, and MI attacks, show MIDRE achieves state-of-the-art privacy-utility trade-offs. Ablation studies and combinations with existing defenses further validate its robustness.
﻿serving machine learning.

**Audience:**

Yes

**Claims And Evidence:**

Yes

**Requested Changes:**

See questions above

**Strengths And Weaknesses:**

Pros
1The method is simple yet highly effective, achieving state-of-the-art performance in privacy-utility trade-offs while enabling seamless integration with existing baseline defenses.

2The authors provide feature-space visualizations to explain why RE works, offering insights into how it disrupts MI attacks by creating discrepancies between private and reconstructed image features.

3The paper includes comprehensive experiments and ablation studies, validating the approach across diverse datasets, architectures, and attack scenarios.

Questions

1. The essence of RE is to prevent the model from observing the entire object information in images during training. This raises a concern: current baseline datasets (e.g., face datasets) have a high proportion of useful information (e.g., 70%+ facial content), so RE retains sufficient information. However, for datasets that useful information is inherently sparse (like object tracking dataset), directly applying RE might lose all useful information or fail to mask partial information effectively. How do the authors address this limitation? In other words, how does the author consider the relationship between the proportion of useful information in the image and the size of RE masking?

2. The RE operation reminds me of the techniques in Masked Autoencoders (MAE)(arXiv:2111.06377). Have the authors explored alternative masking strategies (e.g., smaller but more patches, pixel-level masking)? Such variations might offer better results, particularly for high-resolution or complex data.

In sum, this paper presents a novel and practical defense mechanism against MI attacks, with clear theoretical and empirical contributions. The simplicity of RE, combined with its effectiveness, makes it an attractive solution for privacy-pre

---

> ### Author Response · Authors · 2025-05-26
> **Thank You for the Thoughtful and Supportive Feedback**
>
> We sincerely thank the reviewer for their time and valuable comments.
>
> > 1. The essence of RE is to prevent the model from observing the entire object information in images during training. This raises a concern: current baseline datasets (e.g., face datasets) have a high proportion of useful information (e.g., 70%+ facial content), so RE retains sufficient information. However, for datasets that useful information is inherently sparse (like object tracking dataset), directly applying RE might lose all useful information or fail to mask partial information effectively. How do the authors address this limitation? In other words, how does the author consider the relationship between the proportion of useful information in the image and the size of RE masking?
>
> We appreciate the reviewer’s insightful comment regarding the applicability of RE to datasets where useful information is sparse, such as object tracking datasets. We agree that in such scenarios, indiscriminate masking could risk either removing all useful content or failing to meaningfully obscure sensitive regions.
>
> To address this, we believe that the RE approach remains applicable but requires dataset-specific adjustments. In particular, for object tracking datasets, our strategy would involve first identifying the bounding boxes of private or sensitive objects. The RE masking can then be applied within these bounding boxes, rather than over the entire image. This ensures that masking is targeted and proportional to the relevant regions, preserving sufficient contextual information while still achieving privacy protection. The masking ratio (with similar value to what we have applied in our experiments) would thus be defined relative to the bounding box size—not the full image—to maintain control over how much of the object is obscured.
>
> Moreover, we note that current model inversion research has focused solely on image classification, and to the best of our knowledge, there has been no investigation of model inversion attacks on object tracking datasets. We agree this is an important future direction, and we see value in extending our defense against future object-level model inversion attacks.
>
> > 2. The RE operation reminds me of the techniques in Masked Autoencoders (MAE)(arXiv:2111.06377). Have the authors explored alternative masking strategies (e.g., smaller but more patches, pixel-level masking)? Such variations might offer better results, particularly for high-resolution or complex data.
>
> We thank the reviewer for the valuable suggestion. In response, we have explored two additional masking approaches: (1) masking random pixels, and (2) masking multiple patches. **The results have been included in the updated version of Supp, Section D.1.**
>
> * **Masking random pixels**: Instead of masking a square region as in our proposed Random Erasing (RE) method, we apply masking at the pixel level. For example, we randomly mask 10\% of the image pixels by replacing them with random values. In our experiments, we train the target model with varying levels of random pixel masking, ranging from 10\% ($p = 0.1$) to 60\% ($p = 0.6$).
>
> * **Masking multiple patches**: Instead of masking a single large square region, we apply multiple smaller masks to the image, creating random masking shapes in the training images. In our experiments, we randomly mask each training image with either 4 small patches (4P) or 8 small patches (8P). To ensure a fair comparison with MIDRE, we adjust the patch sizes accordingly. For 4P, we set $\alpha_e\in [0.025, 0.1]$, so that the total area of the four small patches is approximately equivalent to that of MIDRE with  $\alpha_e \in [0.1, 0.4])$ (RE($\alpha_h= 0.4$)). Similarly, for 8P, we use $\alpha_e\in [0.0125, 0.1]$, making the total masked area comparable to MIDRE with $\alpha_e \in [0.1, 0.8]$ (RE($\alpha_h= 0.8$)).
>
> The results show that both masking strategies are less effective than our proposed MIDRE in both lowering attack accuracy and preserving natural accuracy.

---

> > ### Comment · Reviewer_njFH · 2025-06-12
> > **My concerns have been addressed**
> >
> > Thanks for the detailed response, which addresses all of my concerns.

---

### Review · Reviewer_24UE · 2025-05-18

**Summary Of Contributions:**

This paper presents a novel approach to enhancing model robustness against model inversion attacks. Rather than focusing on model architecture or training loss, the authors offer a fresh perspective by intervening directly at the training data level. By randomly erasing regions of input images, the method aims to reduce the amount of private information embedded in the model. The visualization of the embedding space, along with comparative experiments, demonstrates the strong defensive effectiveness of the random erasing strategy.

**Audience:**

Yes

**Broader Impact Concerns:**

NA.

**Claims And Evidence:**

Yes

**Requested Changes:**

In the Introduction, most of the MIAs discussed are from 2023 or earlier. It might be helpful to consider including some more recent studies published in top-tier conferences, if available, to further enhance the comprehensiveness of the paper.

**Strengths And Weaknesses:**

Strengths:
- The existing defenses against model inversion typically target either the architecture or the training loss. This work, however, adopts an alternative perspective by intervening directly on the training data, offering a fresh angle to the ongoing discourse on model inversion defenses.
- The proposed approach is both well-justified and conceptually straightforward. The visualization presented in Figure 3 lends further credibility to the method, illustrating its effectiveness in modifying the embedding space in an intuitive and interpretable manner.
- The experimental evaluation is comprehensive, covering various datasets, model architectures, as well as multiple attack and defense scenarios. The method consistently demonstrates improved performance across most settings, providing solid support for its effectiveness as a defense strategy.

Weaknesses:
- The experimental evaluation thoughtfully considers existing defenses and includes ablations on two key components: mask fraction and location. Nonetheless, it might be worth considering additional baselines to further strengthen the analysis. For instance, exploring alternative masking strategies, such as random pixels, or using varied shapes like rectangles or circles

---

> ### Author Response · Authors · 2025-05-26
> **Thank You for the Thoughtful and Supportive Feedback**
>
> We sincerely thank the reviewer for their time and valuable comments.
> > The experimental evaluation thoughtfully considers existing defenses and includes ablations on two key components: mask fraction and location. Nonetheless, it might be worth considering additional baselines to further strengthen the analysis. For instance, exploring alternative masking strategies, such as random pixels, or using varied shapes like rectangles or circles
>
> We thank the reviewer for the valuable suggestion. In **Section D.4** of our supplementary, we included results using different rectangular masking shapes, controlled by varying the aspect ratio.
>
> We further explore two additional masking approaches in this update: (1) masking random pixels, and (2) masking multiple patches. **The results have been included in the updated version of Supp, Section D.1.**
>
> * **Masking random pixels**: Instead of masking a square region as in our proposed Random Erasing (RE) method, we apply masking at the pixel level. For example, we randomly mask 10\% of the image pixels by replacing them with random values. In our experiments, we train the target model with varying levels of random pixel masking, ranging from 10\% ($p = 0.1$) to 60\% ($p = 0.6$).
>
> * **Masking multiple patches**: Instead of masking a single large square region, we apply multiple smaller masks to the image, creating random masking shapes in the training images. In our experiments, we randomly mask each training image with either 4 small patches (4P) or 8 small patches (8P). To ensure a fair comparison with MIDRE, we adjust the patch sizes accordingly. For 4P, we set $\alpha_e\in [0.025, 0.1]$, so that the total area of the four small patches is approximately equivalent to that of MIDRE with  $\alpha_e \in [0.1, 0.4])$ (RE($\alpha_h= 0.4$)). Similarly, for 8P, we use $\alpha_e\in [0.0125, 0.1]$, making the total masked area comparable to MIDRE with $\alpha_e \in [0.1, 0.8]$ (RE($\alpha_h= 0.8$)).
>
> The results show that both masking strategies are less effective than our proposed MIDRE in both lowering attack accuracy and preserving natural accuracy.
>
>
> > Requested Changes: In the Introduction, most of the MIAs discussed are from 2023 or earlier. It might be helpful to consider including some more recent studies published in top-tier conferences, if available, to further enhance the comprehensiveness of the paper.
>
> Thank you for your helpful suggestion. We agree that incorporating more recent studies will enhance the comprehensiveness of the Introduction. In fact, our experimental evaluation already includes IF-GMI (Qiu et al., 2024), a state-of-the-art MI attack. We will revise the Introduction to update recent works.
>
> Yixiang Qiu, Hao Fang, Hongyao Yu, Bin Chen, MeiKang Qiu, and Shu-Tao Xia. “A closer look at gan priors: Exploiting intermediate features for enhanced model inversion attacks”, In ECCV 2024

---

> > ### Comment · Reviewer_24UE · 2025-06-12
> >
> > Thank you for the detailed response. My concerns are well-addressed.

---

### Review · Reviewer_1fBS · 2025-06-02

**Summary Of Contributions:**

This paper revisits the model-inversion attacks from a data-centric perspective, which is quite novel to the field. The previous defense methods mainly want to train a better model with a given private dataset, yet this paper finds that we can change inputs and then train a model with a common loss function (no additional part like previous methods did).

This contribution is significant in my view because it provides other perspectives to understand and defend against model-inversion attacks.

**Audience:**

Yes

**Claims And Evidence:**

Yes

**Requested Changes:**

Weaknesses:

1. In evaluating the performance of defense MI methods, attack accuracy is only one metric. More metrics should be considered in the experiments, like FID. Note that TMLR mainly requires the proper claims; please do not overclaim the performance on all metrics.

2. The inverted private images should be shown in this paper, which can help understand this research direction better. It is interesting to see what kind of images we can obtain by the proposed defense method.

**Strengths And Weaknesses:**

Strengths:

1. The contribution of this paper is significant. It provides other perspectives to understand and defend against model-inversion attacks, which can motivate more papers in this research direction.

2. The paper is written clearly. Reviewers can easily understand the main contributions after reading the introduction. The method is then introduced very clearly.

3. The experiments show enough evidence to support the basic claim of this paper. The performance, in terms of attack accuracy and accuracy, can well support the claim.

Weaknesses:

1. In evaluating the performance of defense MI methods, attack accuracy is only one metric. More metrics should be considered in the experiments, like FID. Note that TMLR mainly requires the proper claims; please do not overclaim the performance on all metrics.

2. The inverted private images should be shown in this paper, which can help understand this research direction better. It is interesting to see what kind of images we can obtain by the proposed defense method.

---

> ### Author Response · Authors · 2025-06-11
> **Thank you for the helpful feedback**
>
> We sincerely thank the reviewer for their time and valuable comments.
>
> > In evaluating the performance of defense MI methods, attack accuracy is only one metric. More metrics should be considered in the experiments, like FID. Note that TMLR mainly requires the proper claims; please do not overclaim the performance on all metrics.
>
> Thank you for the comment. Beside attack accuracy, we included additional metrics, such as KNN distance, $\delta_{face}$, $\delta_{eval}$, and FID scores, in Tables from B.2 to B.10 (Section B.2, Supplementary).
>
> > The inverted private images should be shown in this paper, which can help understand this research direction better. It is interesting to see what kind of images we can obtain by the proposed defense method.
>
> Thank you for the suggestion. We have updated Section 4.2 to include the reconstructed images. Additional samples can be found in Section B.4 of the Supplementary and our Anonymous GitHub Repository (linked on the first page of the Supplementary).
>
> Overall, our reconstructed images have less similarity to private images compared to those from the no-defense model, evidenced by the lower attack accuracy, demonstrating our defense's effectiveness.

---

> > ### Comment · Reviewer_1fBS · 2025-06-12
> >
> > Thanks! My concerns are addressed well.

---

### Decision · Action_Editor_Tqvi · 2025-07-09

**Recommendation:** Accept as is

**Additional Comments:**

This paper introduces a novel defense approach against Model Inversion (MI) attacks by adopting a data-centric perspective. Unlike most existing defenses that modify the model architecture or training procedure, the authors focus on altering the training data itself to minimize the amount of private information encoded during model learning. This is a refreshing and underexplored direction that has the potential to complement existing model-centric strategies.

For the strength, the existing defenses against model inversion typically target either the architecture or the training loss. This work, however, adopts an alternative perspective by intervening directly on the training data, offering a fresh angle to the ongoing discourse on model inversion defenses. The proposed approach is both well-justified and conceptually straightforward. The visualization presented in Figure 3 lends further credibility to the method, illustrating its effectiveness in modifying the embedding space in an intuitive and interpretable manner. The experimental evaluation is comprehensive, covering various datasets, model architectures, as well as multiple attack and defense scenarios. The method consistently demonstrates improved performance across most settings, providing solid support for its effectiveness as a defense strategy. The authors address reviewers' concerns well in their rebuttal. Thus, I would like to recommend accept as is.

**Audience:**

Yes

**Audience Explanation:**

The contribution of this paper is significant. It provides other perspectives to understand and defend against model-inversion attacks, which can motivate more papers in this research direction. The paper is written clearly. Reviewers can easily understand the main contributions after reading the introduction. The method is then introduced very clearly.

**Claims And Evidence:**

Yes

**Claims Explanation:**

The experiments show enough evidence to support the basic claim of this paper. The performance, in terms of attack accuracy and accuracy, can well support the claim.